# Just Avoid Robust Inaccuracy: Boosting Robustness Without Sacrificing Accuracy

## Abstract

While current methods for training robust deep learning models optimize robust accuracy, they significantly reduce natural accuracy, hindering their adoption in practice. Further, the resulting models are often both robust and inaccurate on numerous samples, providing a false sense of safety for those. In this work, we extend prior works in three main directions. First, we explicitly train the models to jointly maximize robust accuracy and minimize robust inaccuracy. Second, since the resulting models are trained to be robust only if they are accurate, we leverage robustness as a principled abstain mechanism. Finally, this abstain mechanism allows us to combine models in a compositional architecture that significantly boosts overall robustness without sacrificing accuracy. We demonstrate the effectiveness of our approach for empirical and certified robustness on six recent state-of-the-art models and four datasets. For example, on `CIFAR-10` with $\varepsilon_\infty = 1/255$, we successfully enhanced the robust accuracy of a pre-trained model from 26.2% to 87.8% while even slightly increasing its natural accuracy from 97.8% to 98.0%.

## 1 Introduction

In recent years, there has been a significant amount of work that studies and improves adversarial (Carlini & Wagner, 2017; Croce & Hein, 2020b; Goodfellow et al., 2014; Madry et al., 2018; Szegedy et al., 2013) and certified robustness (Balunovic & Vechev, 2019; Cohen et al., 2019; Salman et al., 2019; Xu et al., 2020; Zhai et al., 2020; Zhang et al., 2019b) of neural networks. However, currently, there is a key limitation that hinders the wider adoption of robust models in practice.

**Robustness vs Accuracy Tradeoff**   Despite substantial progress in training robust models, existing robust training methods typically improve model robustness at the cost of decreased standard accuracy. To address this limitation, a number of recent works study this issue in detail and propose new methods to mitigate it (Mueller et al., 2020; Raghunathan et al., 2020; Stutz et al., 2019; Yang et al., 2020).

**Our Work**   In this work, we advance the line of work that aims to boost robustness without sacrificing accuracy, but we approach the problem from a new perspective – by avoiding robust inaccuracy.

Concretely, we propose a new training method that jointly maximizes robust accuracy while minimizing robust inaccuracy. We illustrate the effect of our training on a synthetic dataset (three classes sampled from Gaussian distributions) in Figure 1, showing the decision boundaries of three models, trained using standard training $\mathcal{L}_{std}$, adversarial training $\mathcal{L}_{\text{TRADES}}$ (Zhang et al., 2019a), and our training $\mathcal{L}_{\text{ERA}}$ (Equation 4). First, observe that while the $\mathcal{L}_{std}$ trained model achieves 100% accuracy, only 91.1% of these samples are robust (and accurate). When using $\mathcal{L}_{\text{TRADES}}$, we can observe the

Table 1: Improvement of applying our approach to models trained to optimize natural accuracy only. Here, $\mathcal{R}_{rob}^{acc}$ denotes the robust accuracy and $\mathcal{R}_{nat}$ denotes the standard (non-adversarial) accuracy.

|  | CIFAR-10 Zhao et al. (2020), $\mathcal{B}_{1/255}^{\infty}$ | CIFAR-100 (WideResNet-28-10), $\mathcal{B}_{2/255}^{\infty}$ | MTSD (ResNet-50), $\mathcal{B}_{2/255}^{\infty}$ | SBB (ResNet-50), $\mathcal{B}_{2/255}^{\infty}$ |
|---|---|---|---|---|
| $\mathcal{R}_{rob}^{acc}$ | 26.2 $\xrightarrow{+61.6\%}$ 87.8 | 3.1 $\xrightarrow{+38.8\%}$ 41.9 | 40.7 $\xrightarrow{+29.2\%}$ 69.9 | 44.7 $\xrightarrow{+37.7\%}$ 82.4 |
| $\mathcal{R}_{nat}$ | 97.8 $\xrightarrow{+0.2\%}$ 98.0 | 80.17 $\xrightarrow{+0.01\%}$ 80.18 | 93.8 $\xrightarrow{+0.2\%}$ 94.0 | 91.4 $\xrightarrow{-0.1\%}$ 91.3 |

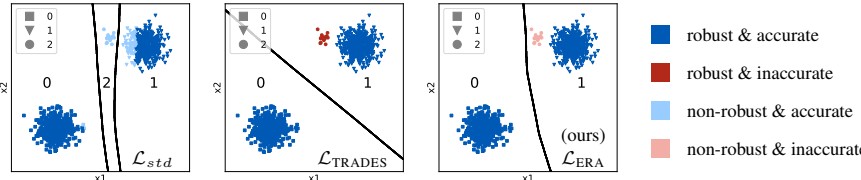

Figure 1: Decision regions for models trained via standard training $\mathcal{L}_{std}$, adversarial training $\mathcal{L}_{\text{TRADES}}$ (Zhang et al., 2019a), and our training $\mathcal{L}_{\text{ERA}}$ (Equation 4). Here, our $\mathcal{L}_{\text{ERA}}$ achieves the same robust accuracy as $\mathcal{L}_{\text{TRADES}}$ but avoids all robust inaccurate samples by making them non-robust. Note that all models predict over all three classes, however, the decision regions for class 2 of the $\mathcal{L}_{\text{TRADES}}$ and $\mathcal{L}_{\text{ERA}}$ trained models are too small to be visible. For more details, please refer to Appendix A.2.

robustness vs accuracy tradeoff – the robust accuracy improves to $98.4\%$ at the expense of $1.6\%$ (robust) inaccuracy. In contrast, using $\mathcal{L}_{\text{ERA}}$, we retain the high robust accuracy of $98.4\%$ but avoid all robust inaccurate samples by appropriately shifting the decision boundary, rendering them non-robust.

Since our models are trained to be robust only if they are accurate, we leverage robustness as a principled abstain mechanism. This abstain mechanism then allows us to combine models in a compositional architecture that significantly boosts overall robustness without sacrificing accuracy. Concretely, in Figure 1, we would define a selector model that abstains on all non-robust samples. Then, the abstained (non-robust) samples are evaluated by the standard trained model $\mathcal{L}_{std}$, while the selected samples are evaluated using the robust model $\mathcal{L}_{\text{ERA}}$. This allows us to achieve the best of both models – high robust accuracy ($98.4\%$), high natural accuracy ($100\%$), and no robust inaccuracy.

We show the practical effectiveness of our approach by instantiating it over several datasets and existing robust models for both empirical and certified robustness. Table 1 summarizes the main results of our approach, showing that we significantly improve the robust accuracy $\mathcal{R}_{rob}^{acc}$ of standard trained non-compositional models, with minimal loss of standard accuracy $\mathcal{R}_{nat}$. In fact, in most of the cases, the compositional architecture even slightly improves the standard accuracy. We release our code at: `https://anonymous.4open.science/r/robust-abstain-09DD`.

## 2 RELATED WORK

There is a growing body of work that extends models with an abstain option. Existing approaches include selection mechanisms such as entropy selection (Mueller et al., 2020), selection function (Cortes et al., 2016; Geifman & El-Yaniv, 2019; Mueller et al., 2020), softmax response (Geifman & El-Yaniv, 2017; Stutz et al., 2020), or explicit abstain class (Laidlaw & Feizi, 2019; Liu et al., 2019). In our work, we explore an alternative selection mechanism that uses model robustness. The advantage of this formulation is that the selector provides strong guarantees for each sample and never produces false-positive selections. The disadvantage is that it introduces a significant runtime overhead, compared to many other methods that require only a single forward pass.

Other recent works address adversarial examples through model calibration. Stutz et al. (2020) proposes biasing models towards low confidence predictions on adversarial examples, which allows rejecting them through a softmax response selector. An alternative approach is taken by Gal & Ghahramani (2018); Kingma et al. (2015); Molchanov et al. (2017), which train Bayesian neural networks to estimate prediction uncertainty by approximating the moments of the posterior predictive distribution, or by Sensoy et al. (2018), which estimates the posterior distribution using a deterministic neural network from data. Instead of calibrating model confidence, in our work, we calibrate model robustness, by optimizing the model towards non-robust predictions on misclassified examples.

Simultaneously, several recent works investigate the robustness and accuracy tradeoff both theoretically (Dobriban et al., 2020; Yang et al., 2020) and practically by proposing new methods to mitigate it. Stutz et al. (2019) considers a new method based on on-manifold adversarial examples, which are more aligned with the true data distribution than the $\ell_p$-norm noise models. Mueller et al. (2020) focuses on deterministic certification and proposes using compositional models to control the robustness and accuracy tradeoff. In our work, we also use compositional models, but we focus on

empirical and probabilistic certified robustness. Our selector formulation is based on a new training that minimizes robust inaccuracy and can be used to fine-tune any existing robust model. Further, we provide individual robustness at inference time, rather than distributional robustness considered in prior works.

Finally, some recent works also consider learning on misclassified examples. For example, MMA (Ding et al., 2018) maximizes the margins of correctly classified examples while minimizing the classification loss on misclassified examples. MART (Wang et al., 2019) combines the standard adversarial risk with a consistency loss that optimizes misclassified examples towards robust predictions. Note, that this formulation actively encourages the model toward robust inaccurate predictions, while our work does the opposite – we minimize robust inaccuracy by penalizing robust misclassified examples.

## 3 PRELIMINARIES

Let $f_\theta \colon \mathbb{R}^d \to \mathbb{R}^k$ be a neural network classifying inputs $\boldsymbol{x} \in \mathcal{X} \subseteq \mathbb{R}^d$ to outputs $\mathbb{R}^k$ (e.g., logits or probabilities). The hard classifier induced by the network is given as $F_\theta(\boldsymbol{x}) = \arg\max_{i \in \mathcal{Y}} f_\theta(\boldsymbol{x})_i$, where $f_\theta(\boldsymbol{x})_i$ is the output for the $i$-th class and $\mathcal{Y}, |\mathcal{Y}| = k$ is the finite set of discrete labels.

**Natural Accuracy**    Given a distribution over input-label pairs $\mathcal{D}$ and a classifier $F_\theta \colon \mathcal{X} \to \mathcal{Y}$, an input-label pair $(\boldsymbol{x}, y)$ is considered accurate iff the classifier $F_\theta$ predicts the correct label $y$ for $\boldsymbol{x}$:

$$\mathcal{R}_{nat}(F_\theta) = \mathbb{E}_{(\boldsymbol{x},y)\sim\mathcal{D}} \quad \mathbf{1}\{F_\theta(\boldsymbol{x}) = y\}$$

**Robust Accuracy**    Given an input-label pair $(\boldsymbol{x}, y)$, we say that the classifier $F_\theta$ is robust and accurate iff it predicts the correct label $y$ for all samples from a predefined region $\mathcal{B}_\varepsilon^p(\boldsymbol{x})$, such as a $\ell_p$-norm ball centered at $\boldsymbol{x}$ with radius $\varepsilon$, i.e., $\mathcal{B}_\varepsilon^p(\boldsymbol{x}) := \{\boldsymbol{x}' \colon ||\boldsymbol{x}' - \boldsymbol{x}||_p \le \varepsilon\}$. Formally:

$$\mathcal{R}_{rob}^{acc}(F_\theta) = \mathbb{E}_{(\boldsymbol{x},y)\sim\mathcal{D}} \quad \mathbf{1}\{F_\theta(\boldsymbol{x}) = y\} \wedge \mathbf{1}\{\forall \boldsymbol{x}' \in \mathcal{B}_\varepsilon^p(\boldsymbol{x}). \, F_\theta(\boldsymbol{x}') = F_\theta(\boldsymbol{x})\} \tag{1}$$

**Robust Inaccuracy**    Similarly to robust accuracy, an input-label pair $(\boldsymbol{x}, y)$ is considered robustly inaccurate iff the classifier $F_\theta$ predicts an incorrect label $F_\theta(\boldsymbol{x}) \neq y$ and $F_\theta$ is robust towards that misprediction for all inputs in $\mathcal{B}_\varepsilon^p(\boldsymbol{x})$. Formally, the robust inaccuracy is defined as:

$$\mathcal{R}_{rob}^{\neg acc}(F_\theta) = \mathbb{E}_{(\boldsymbol{x},y)\sim\mathcal{D}} \quad \mathbf{1}\{F_\theta(\boldsymbol{x}) \neq y\} \wedge \mathbf{1}\{\forall \boldsymbol{x}' \in \mathcal{B}_\varepsilon^p(\boldsymbol{x}). \, F_\theta(\boldsymbol{x}') = F_\theta(\boldsymbol{x})\} \tag{2}$$

## 4 REDUCING ROBUST INACCURACY

In this section, we present our training method that extends existing robust training approaches by also considering samples that are robust but inaccurate. We start by describing a high-level problem statement which we then instantiate for both empirical robustness as well as certified robustness.

**Problem Statement**    Given a distribution over input-label pairs $\mathcal{D}$, our goal is to find model parameters $\theta$ such that the resulting model maximizes robust accuracy, while at the same time minimizing robust inaccuracy. Concretely, this translates to the following optimization objective:

$$\arg\min_\theta \mathbb{E}_{(\boldsymbol{x},y)\sim\mathcal{D}} \quad \underbrace{\beta \cdot \mathcal{L}_{rob}(\boldsymbol{x}, y)}_{\text{optimize robust accuracy}} \quad + \quad \underbrace{\mathbf{1}\{F_\theta(\boldsymbol{x}) \neq y\} \cdot \mathcal{L}_{rob}^{\neg acc}(\boldsymbol{x}, y)}_{\text{penalize robust inaccuracy}} \tag{3}$$

where $\beta \in \mathbb{R}^+$ is a regularization term, $\mathbf{1}\{F_\theta(\boldsymbol{x}) \neq y\}$ is an indicator function denoting samples for which the model is inaccurate, and $\mathcal{L}_{rob}(\boldsymbol{x}, y)$ with $\mathcal{L}_{rob}^{\neg acc}(\boldsymbol{x}, y)$ are loss functions that optimize robust accuracy and penalize robust inaccuracy, respectively. Here, the first loss function $\mathcal{L}_{rob}(\boldsymbol{x}, y)$ is standard and can be directly instantiated using existing approaches. The main challenge comes in defining the second loss term, as well as ensuring that the resulting formulation is easy to optimize, e.g., by defining a smooth approximation of the non-differentiable indicator function.

### 4.1 ADVERSARIAL TRAINING

We instantiate the loss function from Equation 3 when training empirically robust models as follows:

$$\mathcal{L}_{\text{ERA}} = \beta \cdot \mathcal{L}_{\text{TRADES}}(f_\theta, (\boldsymbol{x}, y)) + (1 - f_\theta(\boldsymbol{x})_y) \min_{\boldsymbol{x}' \in \mathcal{B}_\varepsilon^p(\boldsymbol{x})} \ell_{\text{CE}}(f_\theta(\boldsymbol{x}'), \arg\max_{c \in \mathcal{Y} \setminus \{F_\theta(\boldsymbol{x})\}} f_\theta(\boldsymbol{x}')_c) \tag{4}$$

Following, we introduce each term in more detail and discuss the motivation behind our formulation.

$\mathcal{L}_{rob}$     To instantiate $\mathcal{L}_{rob}$, we can use any existing adversarial training method (Ding et al., 2018; Goodfellow et al., 2014; Wang et al., 2019; Zhang et al., 2019a). For example, considering TRADES (Zhang et al., 2019a), $\mathcal{L}_{rob}$ is instantiated as:

$$\mathcal{L}_{\text{TRADES}} := \ell_{\text{CE}}(f_\theta(\boldsymbol{x}), y) + \gamma \max_{\boldsymbol{x}' \in \mathcal{B}_\varepsilon^p(\boldsymbol{x})} D_{\text{KL}}(f_\theta(\boldsymbol{x}), f_\theta(\boldsymbol{x}')) \tag{5}$$

where $D_{\text{KL}}$ is the Kullback-Leibler divergence (Kullback & Leibler, 1951).

$\mathbf{1}\{F_\theta(\boldsymbol{x}) \neq y\}$     Next, we consider the indicator function, which encourages learning on inaccurate samples. Since the indicator function is computationally intractable, we replace the hard qualifier by a soft qualifier $1 - f_\theta(\boldsymbol{x})_y$. The soft qualifier will be small for accurate and large for inaccurate samples, thus providing a smooth approximation of the original indicator function.

$\mathcal{L}_{rob}^{\neg acc}$     Third, we define the loss that penalizes robust but inaccurate samples. This can be formulated similar to the adversarial training objective (Madry et al., 2018), however, instead of optimizing the prediction of the adversarial example $f_\theta(\boldsymbol{x}')$ towards the correct label $y$, we optimize towards the most likely adversarial label $\arg\max_{c \in \mathcal{Y} \setminus \{F_\theta(\boldsymbol{x})\}} f_\theta(\boldsymbol{x}')_c$. This leads to the following formulation:

$$\min_{\boldsymbol{x}' \in \mathcal{B}_\varepsilon^p(\boldsymbol{x})} \ell_{\text{CE}}(f_\theta(\boldsymbol{x}'), \arg\max_{c \in \mathcal{Y} \setminus \{F_\theta(\boldsymbol{x})\}} f_\theta(\boldsymbol{x}')_c) \tag{6}$$

The purpose of the $\mathcal{L}_{rob}^{\neg acc}$ loss is to penalize robustness by making the model non-robust. As a result, it is sufficient to consider only a single non-robust example, thus the minimization (rather than maximization) in the loss objective[1].

## 4.2   CERTIFIED TRAINING

Similarly to Section 4.1, we now instantiate the loss function from Equation 3 for probabilistic certified robustness via randomized smoothing (Cohen et al., 2019). Randomized smoothing constructs a smoothed classifier $G_\theta \colon \mathcal{X} \to \mathcal{Y}$ from a base classifier $F_\theta$, where $G_\theta(\boldsymbol{x})$ predicts the class which $F_\theta$ is most likely to return when $\boldsymbol{x}$ is perturbed under isotropic Gaussian noise. Our proposed instantiation of Equation 3 for probabilistic certified robustness is as follows:

$$\mathcal{L}_{\text{CRA}}(f_\theta, (\boldsymbol{x}, y)) = \beta \cdot \mathcal{L}_{noise}(f_\theta, (\boldsymbol{x}, y)) + \frac{1}{k} \sum_{j=1}^{k} \left(1 - f_\theta(\boldsymbol{x} + \boldsymbol{\eta}_j)_y\right) CR(f_\theta, (\boldsymbol{x}, y)) \tag{7}$$

where $\boldsymbol{\eta}_1, ..., \boldsymbol{\eta}_k$ are $k$ *i.i.d.* samples from $\mathcal{N}(0, \sigma^2 \boldsymbol{I})$. Note that, since the robustness guarantees provided by randomized smoothing hold for the smoothed classifier $G_\theta$, the three loss components from Equation 3 need to be formulated with respect to the smoothed classifier $G_\theta$.

$\mathcal{L}_{rob}$     To instantiate $\mathcal{L}_{rob}$, we can use any existing certified training method for randomized smoothing, such as the methods defined by Cohen et al. (2019) or Zhai et al. (2020). Concretely, when using Cohen et al. (2019), the loss is defined using Gaussian noise augmentation:

$$\mathcal{L}_{noise} := \ell_{\text{CE}}(f_\theta(\boldsymbol{x} + \boldsymbol{\eta}), y), \quad \boldsymbol{\eta} \sim \mathcal{N}(0, \sigma^2 \boldsymbol{I}) \tag{8}$$

$\mathbf{1}\{F_\theta(\boldsymbol{x}) \neq y\}$     We again replace the computationally intractable hard qualifier by a soft qualifier $\mathbb{E}_{\boldsymbol{\delta} \sim \mathcal{N}(0, \sigma^2 \boldsymbol{I})}[1 - f_\theta(\boldsymbol{x} + \boldsymbol{\delta})_y]$, which encodes the misprediction probability of the smoothed classifier. In practice, we approximate expectations over Gaussians via Monte Carlo sampling, thus leading to the approximated soft inaccuracy qualifier $1/k \sum_{j=1}^{k} 1 - f_\theta(\boldsymbol{x} + \boldsymbol{\eta}_j)_y$.

$\mathcal{L}_{rob}^{\neg acc}$     Finally, we instantiate the $\mathcal{L}_{rob}^{\neg acc}$ loss term, which encourages the model toward non-robust predictions on robust but inaccurate samples. We propose to minimize robustness by directly minimizing the certified radius of the smoothed classifier $G_\theta$. The certified radius formulation by

---

[1]Naturally, this assumes that the method used to check robustness can correctly detect the non-robustness, even if it is caused by a single example. Note that, for a fair evaluation, we use a relatively weak 10-step PGD (Madry et al., 2018) attack during training and a strong 40-step APGD (Croce & Hein, 2020b) for evaluation.

Cohen et al. (2019) involves a sum of indicator functions, which is not differentiable. However, Zhai et al. (2020) have recently proposed the following differentiable certified radius formulation:

$$CR(f_\theta, (\boldsymbol{x}, y)) = \frac{\sigma}{2}\big[\Phi^{-1}\big(\frac{1}{k}\sum_{j=1}^{k} f_\theta(\boldsymbol{x} + \boldsymbol{\eta}_j; \Gamma)_y\big) - \Phi^{-1}\big(\max_{y'\neq y}\frac{1}{k}\sum_{j=1}^{k} f_\theta(\boldsymbol{x} + \boldsymbol{\eta}_j; \Gamma)_{y'}\big)\big] \quad (9)$$

where $\Phi^{-1}$ is the inverse of the standard Gaussian CDF, $\Gamma$ is the inverse softmax temperature multiplied with the logits of $f_\theta$, and $\boldsymbol{\eta}_{1:k}$ are $k$ *i.i.d.* samples from $\mathcal{N}(0, \sigma^2 \boldsymbol{I})$. Note that, by setting the loss term $\mathcal{L}_{rob}^{\neg acc}$ to $CR(f_\theta, (\boldsymbol{x}, y))$, we directly penalize robustness of the smoothed classifier $G_\theta$.

## 5 ROBUST ABSTAIN MODELS

Next, we extend the models trained so far by leveraging robustness as a principled abstain mechanism.

**Abstain Model**    Given input space $\mathcal{X} \subseteq \mathbb{R}^d$ and label space $\mathcal{Y}$, a model with an abstain option (El-Yaniv et al., 2010) is a pair of functions $(F_\theta, S)$, where $F_\theta\colon \mathcal{X} \to \mathcal{Y}$ is a classifier and $S\colon \mathcal{X} \to \{0, 1\}$ is a binary selector for $F_\theta$. Let $S(\boldsymbol{x}) = 0$ indicate that the model abstains on input $\boldsymbol{x} \in \mathcal{X}$, while $S(\boldsymbol{x}) = 1$ indicates that the model commits to the classifier $F_\theta$ for input $\boldsymbol{x}$ and predicts $F_\theta(\boldsymbol{x})$.

**Robustness Indicator Selector**    We instantiate abstain models with a robustness indicator selector, that abstains on all non-robust samples. For adversarial robustness, the selector is defined as:

$$S_{\texttt{ERI}}(\boldsymbol{x}) = \mathbf{1}\{\forall \boldsymbol{x}' \in \mathcal{B}(\boldsymbol{x})\colon F_\theta(\boldsymbol{x}') = F_\theta(\boldsymbol{x})\} \quad (10)$$

For certified robustness, the selector is defined as:

$$S_{\texttt{CRI}}(\boldsymbol{x}) = \mathbf{1}\{\forall \boldsymbol{x}' \in \mathcal{B}(\boldsymbol{x})\colon G_\theta(\boldsymbol{x}') = G_\theta(\boldsymbol{x})\} \quad (11)$$

**Robustness Guarantees: Robust Selection**    Similar to robust accuracy, the robustness of an abstain model needs to be evaluated with respect to a threat model. In our work, we consider the same threat model as for the underlying model $F_\theta$, namely $\mathcal{B}_\varepsilon^p(\boldsymbol{x}) \coloneqq \{\boldsymbol{x}'\colon \|\boldsymbol{x}' - \boldsymbol{x}\|_p \leq \varepsilon\}$, a $\ell_p$-norm ball centered at $\boldsymbol{x}$ with radius $\varepsilon$. Then, we define the robust selection of an abstain model as follows:

$$\mathcal{R}_{rob}^{sel}(S) = \mathbb{E}_{(\boldsymbol{x},y)\sim\mathcal{D}} \quad \mathbf{1}\{\forall \boldsymbol{x}' \in \mathcal{B}_\varepsilon^p(\boldsymbol{x}). S(\boldsymbol{x}') = 1\}$$

That is, we say that a model robustly selects $\boldsymbol{x}$ if the selector $S$ would select all valid perturbations $\boldsymbol{x}' \in \mathcal{B}_\varepsilon^p(\boldsymbol{x})$. Combined with our definition of $S_{\texttt{ERI}}$, we obtain the following criterion (cf. Appendix A.3):

$$\mathcal{R}_{rob}^{sel}(S_{\texttt{ERI}}) = \mathbb{E}_{(\boldsymbol{x},y)\sim\mathcal{D}}\mathbf{1}\{\forall \boldsymbol{x}' \in \mathcal{B}_{2\cdot\varepsilon}^p(\boldsymbol{x}). F_\theta(\boldsymbol{x}') = F_\theta(\boldsymbol{x})\}$$

In other words, to guarantee that the selector $S_{\texttt{ERI}}$ is robust for all $\boldsymbol{x}' \in \mathcal{B}_\varepsilon^p(\boldsymbol{x})$, we in fact need to check robustness of the model $F_\theta$ to double that region $\boldsymbol{x}' \in \mathcal{B}_{2\cdot\varepsilon}^p(\boldsymbol{x})$. This is important in order to obtain the correct guarantees and is reflected in our evaluation in Section 7.

Note that when evaluating robust selection for certified training, it is sufficient to show that the smoothed model $G_\theta$ can be certified with a radius $R \geq \varepsilon$. Then, the smoothed model guarantees that $G_\theta(\boldsymbol{x}') = c_A$ for all $\boldsymbol{x}' \in \mathcal{B}_\varepsilon^p(\boldsymbol{x})$, which is equivalent to our condition $\forall \boldsymbol{x}' \in \mathcal{B}(\boldsymbol{x})\colon G_\theta(\boldsymbol{x}') = G_\theta(\boldsymbol{x})$.

## 6 BOOSTING ROBUSTNESS WITHOUT ACCURACY LOSS

Consider an abstain model $(F_\theta, S)$ and a dataset $\mathcal{D}$. The selector $S$ partitions $\mathcal{D}$ into two disjoint subsets – the abstained inputs $\mathcal{D}_{\neg s}$ and the selected inputs $\mathcal{D}_s$ for which $F_\theta$ makes a prediction. For some tasks, making a best-effort prediction on all samples $\mathcal{D}_s \cup \mathcal{D}_{\neg s}$ may be desirable, which leads to compositional architectures, already used by prior works (Mueller et al., 2020; Wong et al., 2018).

Let $H = ((F_{robust}, S), F_{core})$ be a 2-compositional architecture consisting of a selection mechanism $S$, a robustly trained model $F_{robust}$, and a core model $F_{core}$. Given an input $\boldsymbol{x} \in \mathcal{X}$, the selector $S$ decides whether the model is confident on $\boldsymbol{x}$ and commits to the robust model $F_{robust}$ or whether the model should abstain and fall back to the core model $F_{core}$. Formally:

$$H(\boldsymbol{x}) = S(\boldsymbol{x}) \cdot F_{robust}(\boldsymbol{x}) + (1 - S(\boldsymbol{x})) \cdot F_{core}(\boldsymbol{x}) \quad (12)$$

While $F_{robust}, F_{core}$ can be chosen arbitrarily, we here combine robust trained models (which have lower natural accuracy), with models trained using standard training (which have high natural accuracy but low robustness). The performance of $H$ then depends on the quality of the selector $S$.

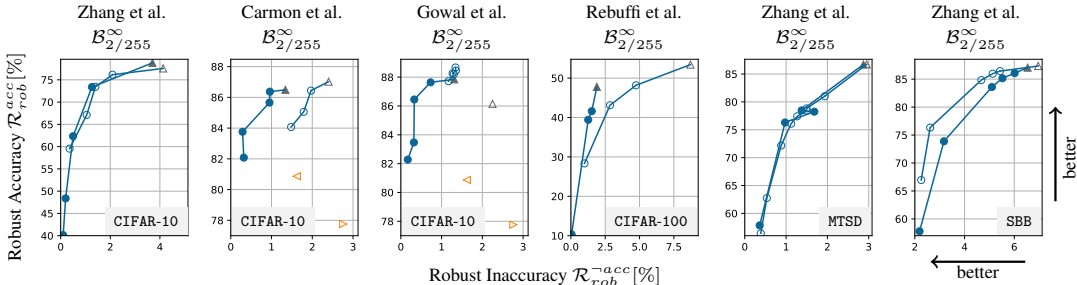

Figure 2: Robust accuracy ($\mathcal{R}_{rob}^{acc}$) and robust inaccuracy ($\mathcal{R}_{rob}^{\neg acc}$) of existing robust models (△, ▲, ◁, ▷), and models fine-tuned with our loss (○, ●). Our approach consistently reduces robust inaccuracy across various datasets, existing models and different regularization levels $\beta$.

## 7 EVALUATION

In this section, we present a thorough evaluation of our approach by instantiating it to four different datasets, six recent state-of-the-art models, for both adversarial and certified robustness, including top-trained models from RobustBench (Croce et al., 2020). We show the following key results:

- Fine-tuning models with our proposed loss successfully decreases robust inaccuracy and provides a Pareto front of models with different robustness tradeoffs.
- Combining our proposed loss and robustness as an abstain mechanism leads to higher robust selection and accuracy compared to softmax response and selection network baselines.
- Our 2-compositional models significantly improve robustness by up to $+61\%$ and slightly increase the natural accuracy by up to $+0.2\%$ (for $\mathcal{B}_{1/255}^{\infty}$ and $\mathcal{B}_{2/255}^{\infty}$).

We perform all experiments on a single GeForce RTX 3090 GPU and use PyTorch (Paszke et al., 2019) for our implementation. The hyperparameters used for our experiments are provided in Appendix A.2.

**Models** Our proposed training method requires neither retraining classifiers from scratch nor modifications to existing classifiers, thus our approach can be applied to fine-tune a wide range of existing models[2]. To demonstrate this, we use the following robust pre-trained models:

For *empirical robustness*, we evaluate existing models from Carmon et al. (2019), Gowal et al. (2020), Rebuffi et al. (2021), and Zhang et al. (2019a), which were all trained for $\varepsilon_{\infty} = 8/255$, and all but the last model are top models in RobustBench (Croce et al., 2020). In our evaluation, we fine-tune each model for 50 epochs for the considered threat model ($\varepsilon_{\infty} \in \{1/255, 2/255, 4/255\}$), using $\mathcal{L}_{\text{TRADES}}$ (Zhang et al., 2019a) and $\mathcal{L}_{\text{ERA}}$ (ours). Further, we also consider models by Ding et al. (2018) and Wang et al. (2019) as additional baselines.

For *certified robustness*, we use a $\sigma = 0.12$ Gaussian noise augmentation trained model by Cohen et al. (2019) and a $\varepsilon_2 = 0.5$ adversarially trained model by Sehwag et al. (2021). Similar to empirical robustness, we fine-tune the models for 50 epochs using $\mathcal{L}_{noise}$ (Cohen et al., 2019) and $\mathcal{L}_{\text{CRA}}$ (ours).

**Datasets** We evaluate our approach on two academic datasets – CIFAR-10 and CIFAR-100 (Krizhevsky et al., 2009), and two commercial datasets – Mapillary Traffic Sign Dataset (MTSD) (Ertler et al., 2020) and a Rail Defect Dataset kindly provided by Swiss Federal Railways (SBB). Consider Appendix A.1 for full details.

When training on the CIFAR-10 and CIFAR-100 datasets, we use the AutoAugment (AA) policy by Cubuk et al. (2018) as the image augmentation. For the MTSD and SBB datasets, we use standard image augmentations (SA) consisting of random cropping, color jitter, and random translation and rotation. For completeness, our evaluation also includes models trained without any data augmentations.

**Metrics** We use the natural accuracy, robust accuracy, and robust inaccuracy as our main evaluation metrics, as defined in Section 3, but evaluated on the corresponding test dataset.

---

[2]Our method can also be used to train from scratch, in which case a scheduler for $\beta$ should be introduced.

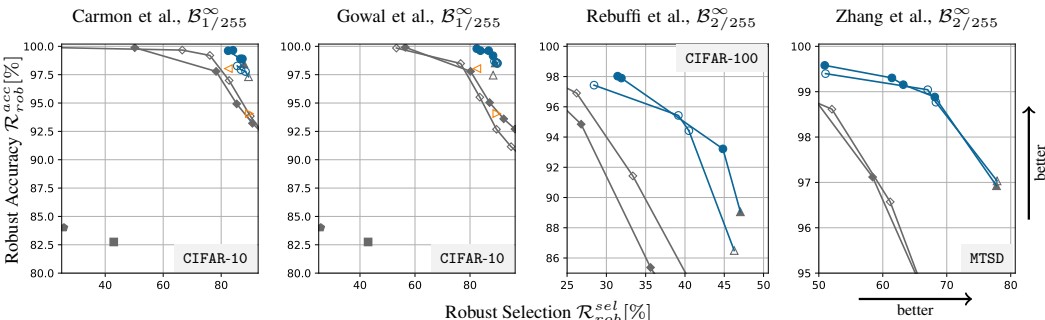

Figure 3: Comparison of different abstain approaches including existing robust classifiers TRADES$_{\text{RI}}$ ($\triangle$, $\blacktriangle$), MART$_{\text{RI}}$ ($\triangleleft$), MMA$_{\text{RI}}$ ($\triangleright$), classifiers fine-tuned with our proposed loss ERA$_{\text{RI}}$ ($\circ$, $\bullet$), selection network ($\blacksquare$, $\spadesuit$), and softmax response ($\diamond$, $\blacklozenge$) abstain models. Higher $\mathcal{R}_{rob}^{sel}$ and $\mathcal{R}_{rob}^{acc}$ is better (top right corner is optimal). Observe that the Parent front of our approach ($\circ$, $\bullet$) generally dominates the results of all baselines, significantly improving robust selection and robust accuracy.

When evaluating empirical robustness, we use 40-step APGD$_{\text{CE}}$ (Croce & Hein, 2020b) (referred to as APGD) to evaluate robustness of classifiers $F_\theta$. To evaluate certified robustness, we use the Monte Carlo algorithm for randomized smoothing from Cohen et al. (2019). We certify 500 test samples and use the same randomized smoothing hyperparameters as Cohen et al. (2019) (cf. Appendix A.2).

## 7.1 REDUCING ROBUST INACCURACY

We first summarize the main results obtained by using our proposed loss functions $\mathcal{L}_{\text{ERA}}$ and $\mathcal{L}_{\text{CRA}}$.

**Empirical Robustness** The results in Figure 2 show the robust accuracy ($\mathcal{R}_{rob}^{acc}$) and robust in-accuracy ($\mathcal{R}_{rob}^{\neg acc}$) of different existing robust models fine-tuned via TRADES (Zhang et al., 2019a) with ($\blacktriangle$) and without ($\triangle$) data augmentations, and the same models fine-tuned via our $\mathcal{L}_{\text{ERA}}$ with ($\bullet$) and without ($\circ$) data augmentations. Further, we show MART (Wang et al., 2019) ($\triangleleft$), and MMA (Ding et al., 2018) ($\triangleright$) finetuned models as additional baselines. We can see that our approach consistently improves over existing models. For example, for CIFAR-10 and $\mathcal{B}_{2/255}^\infty$, the Carmon et al. (2019) model achieves 86.5% robust accuracy, but also 1.34% robust inaccuracy. In contrast, using $\mathcal{L}_{\text{ERA}}$, we obtain a number of models that reduce robust inaccuracy to 0.29%, while still achieving 83.8% robustness. Similar results are obtained for other models, perturbation regions, and datasets (cf. Appendix A.7). We observe that our approach achieves consistently lower robust inaccuracy compared to adversarial training. Further, by varying the regularization term $\beta$, we obtain a Pareto front of optimal solutions.

**Certified Robustness** Similarly, we evaluate the robust accuracy ($\mathcal{R}_{rob}^{acc}$) and robust inaccuracy ($\mathcal{R}_{rob}^{\neg acc}$) for certifiably robust models fine-tuned using $\mathcal{L}_{noise}$ and $\mathcal{L}_{\text{CRA}}$ (ours). In Table 2a, we show results on CIFAR-10 for $\mathcal{B}_{0.12}^2$ and $\mathcal{B}_{0.25}^2$. We observe that our approach achieves lower robust inaccuracy compared to existing models. For example, on CIFAR-10 and $\mathcal{B}_{0.25}^2$, the Cohen et al. (2019) model achieves 62% robust accuracy, but also 1% robust inaccuracy. In contrast, our approach reduces robust inaccuracy to 0.4% while still achieving 53.8% robust accuracy. For the Sehwag et al. (2021) model, our approach even improves both robust accuracy and robust inaccuracy. For $\mathcal{B}_{0.25}^2$, our approach improves the robust accuracy by $+4.8\%$ and reduces the robust inaccuracy by $-0.6\%$.

## 7.2 USING ROBUSTNESS TO ABSTAIN

Next, we evaluate using robustness as an abstain mechanism (Section 5) and how it benefits from the training proposed in our work. We compare the following abstain mechanisms:

*Softmax Response (SR)* (Geifman & El-Yaniv, 2017), which abstains if the maximum softmax output of the model $f_\theta$ is below a threshold $\tau$ for some input $x' \in \mathcal{B}_\varepsilon^p(x)$, that is:

$$S_{\text{SR}}(x) = \mathbf{1}\{\forall x' \in \mathcal{B}_\varepsilon^p(x): \max_{c \in \mathcal{Y}} f_\theta(x')_c \geq \tau\} \tag{13}$$

Similar to $S_{\text{RI}}$, to guarantee robustness of $S_{\text{SR}}$, we need to check the maximum softmax output of $f_\theta$ on double the region $\mathcal{B}^p_{2 \cdot \varepsilon}(\boldsymbol{x})$. To evaluate robustness of $S_{\text{SR}}$, we use a modified version of APGD called APGDconf (Appendix A.5). For each considered model (e.g., Carmon et al. (2019)), we evaluate its corresponding abstain selector: (◇, ◆) CARMON$_{\text{SR}}$, GOWAL$_{\text{SR}}$, etc. (all fine-tuned using TRADES).

*Robustness Indicator (RI) (our work)*, which abstains if the model $F_\theta$ is non-robust:

$$S_{\text{RI}}(\boldsymbol{x}) = \mathbf{1}\{\forall \boldsymbol{x}' \in \mathcal{B}^p_\varepsilon(\boldsymbol{x})\colon F_\theta(\boldsymbol{x}') = F_\theta(\boldsymbol{x})\} \tag{14}$$

Note that, unlike other selectors, our robustness indicator is by design robust against an adversary using the same threat model. For each base model, we consider two instantiations (△, ▲) TRADES$_{\text{RI}}$, and (○, ●) ERA$_{\text{RI}}$ (Equation 4). Further, for CIFAR-10, we also instantiate models from Ding et al. (2018); Wang et al. (2019) with robustness indicator abstain: MART$_{\text{RI}}$ (◄), and MMA$_{\text{RI}}$ (►).

*Selection Network (SN)*, which trains a separate neural network $s_\theta \colon \mathcal{X} \to \mathbb{R}$ and selects if:

$$S_{\text{SN}}(\boldsymbol{x}) = \mathbf{1}\{s_\theta(\boldsymbol{x}) \geq \tau\} \tag{15}$$

When evaluating the robustness of an abstain model $(F_\theta, S_{\text{SN}})$, the robustness of both the classifier and the selection network have to be considered. We compare against two instantiations of this approach, both trained using certified training: (■) ACE-COLT$_{\text{SN}}$ (Balunovic & Vechev, 2019; Mueller et al., 2020), and (⬣) ACE-IBP$_{\text{SN}}$ (Gowal et al., 2018; Mueller et al., 2020).

**Empirical Robustness**  In Figure 3, we compare different abstain approaches using two metrics – robust selection ($\mathcal{R}^{sel}_{rob}$), and the ratio of non-abstained samples that are robust and accurate ($\mathcal{R}^{acc}_{rob}$). We would like to maximize both, but typically there is a tradeoff between the two. This is evident in Figure 3, where both our approach and softmax response produce a Pareto front of optimal solutions.

Overall, the main results in Figure 3 show that, as designed, our approach consistently improves robust accuracy. For example, on CIFAR-10, $\mathcal{B}^\infty_{1/255}$ and Carmon et al. (2019) model, we successfully improve robust accuracy by +1.18% at the cost of -3.78% decreased robust selection. This is close to optimal since increasing robust accuracy is typically achieved by correctly abstaining on misclassified samples. Interestingly, in some cases, we strictly improve over baseline models by increasing robust accuracy and robust selection. For CIFAR-10, $\mathcal{B}^\infty_{1/255}$, and Gowal et al. (2020) model, we increase robust accuracy by +1.06% and robust selection by +1.61% (training without data augmentations).

Compared to the other abstain methods, our approach generally improves both metrics while also providing much stronger guarantees. Concretely, our approach guarantees that selected samples are robust in the considered threat model. Softmax response only guarantees that all samples in the considered threat model have high confidence and is thus vulnerable to high confidence adversarial examples, and the selection network provides no guarantees with regards to the selector's robustness.

**Certified Robustness**  Applying our training for certified robustness $\mathcal{L}_{\text{CRA}}$ with $\beta = 1.0$ consistently improves robust accuracy $\mathcal{R}^{acc}_{rob}$ of robustness indicator abstain models. In Table 2b, we show our results on CIFAR-10 for $\mathcal{B}^2_{0.12}$ and $\mathcal{B}^2_{0.25}$. For instance, for the Cohen et al. (2019) model trained at $\sigma = 0.12$, we are able to improve the robust accuracy by +0.85% for $\mathcal{B}^2_{0.25}$, at the expense of $-8.8\%$ decrease in robust selection. For the Sehwag et al. (2021) model, our approach improves on both metrics. For $\mathcal{B}^2_{0.25}$, we increase robust accuracy by +0.82% and robust selection by +4.2%.

Table 2: Comparison of existing robust models fine-tuned with $\mathcal{L}_{noise}$ and $\mathcal{L}_{\text{CRA}}$ (ours). Table 2a shows robust accuracy ($\mathcal{R}^{acc}_{rob}$) and robust inaccuracy ($\mathcal{R}^{\neg acc}_{rob}$) of fine-tuned models $F_\theta$. Table 2b shows robust selection ($\mathcal{R}^{sel}_{rob}$) and robust accuracy ($\mathcal{R}^{acc}_{rob}$) of corresponding RI abstain models ($F_\theta, S_{\text{CRI}}$).

| (a) Finetuned models $F_\theta$. | | | | | | (b) Finetuned abstain models $(F_\theta, S_{\text{CRI}})$. | | | |
|---|---|---|---|---|---|---|---|---|---|
| CIFAR-10 | | $\mathcal{B}^2_{0.12}(\sigma=0.6)$ | | $\mathcal{B}^2_{0.25}(\sigma=0.12)$ | | $\mathcal{B}^2_{0.12}(\sigma=0.6)$ | | $\mathcal{B}^2_{0.25}(\sigma=0.12)$ | |
| Model | Finetuning | $\mathcal{R}^{acc}_{rob}$ | $\mathcal{R}^{\neg acc}_{rob}$ | $\mathcal{R}^{acc}_{rob}$ | $\mathcal{R}^{\neg acc}_{rob}$ | $\mathcal{R}^{sel}_{rob}$ | $\mathcal{R}^{acc}_{rob}$ | $\mathcal{R}^{sel}_{rob}$ | $\mathcal{R}^{acc}_{rob}$ |
| Cohen et al. | $\mathcal{L}_{noise}$ | **74.0** | 2.8 | **62.0** | 1.0 | **76.8** | 96.35 | **63.0** | 98.41 |
| | $\mathcal{L}_{\text{CRA}}$ | 71.6 | **2.6** | 53.8 | **0.4** | 74.2 | **96.50** | 54.2 | **99.26** |
| Sehwag et al. | $\mathcal{L}_{noise}$ | 87.0 | 1.8 | 77.4 | 1.4 | 88.8 | 97.97 | 78.8 | 98.22 |
| | $\mathcal{L}_{\text{CRA}}$ | **90.8** | **1.6** | **82.2** | **0.8** | **92.4** | **98.27** | **83.0** | **99.04** |

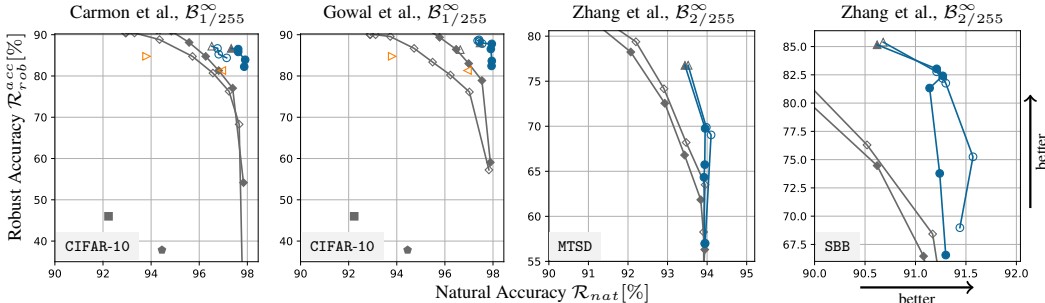

Figure 4: 2-compositional natural ($\mathcal{R}_{nat}$) and robust accuracy ($\mathcal{R}_{rob}^{acc}$) for ERA$_{RI}$ (○, ●), TRADES$_{RI}$ (△, ▲), MART$_{RI}$ (◁), MMA$_{RI}$ (▷), ACE-COLT$_{SN}$, ACE-IBP$_{SN}$ (■, ⬠), and TRADES$_{SR}$ (◇, ◆) models. The core models used in the compositional architectures are listed in Appendix A.10. We can see that the Parent front of our method strictly improves over the prior work in the most important region – significantly improving model robustness while the model accuracy does not decrease.

### 7.3 BOOSTING ROBUSTNESS WITHOUT ACCURACY LOSS

Finally, we present the results of combining the abstain models trained so far with state-of-the-art models trained to achieve high accuracy. Note that, as discussed in Section 5, when evaluating adversarial robustness for $\mathcal{B}_\varepsilon^p$, we in fact need to consider $\mathcal{B}_{2\cdot\varepsilon}^p$ robustness of the abstain model.

A summary of the results is shown in Figure 4. Similar to the results shown so far, the 2-compositional architectures that use models trained by our method (○, ●) improve over existing methods that optimize robust accuracy (△, ▲, ◁, ▷), as well as over models using softmax response (◇, ◆) or selection network (■, ⬠) to abstain. For example, for CIFAR-10 with $\varepsilon_\infty = 1/255$ and the Carmon et al. (2019) model, we improve natural accuracy by +0.58% and +0.62%, while decreasing the robustness only by -2.75% and -2.82%, when training with and without data augmentations respectively.

More importantly, our approach significantly improves robustness of highly accurate non-compositional models, with minimal loss of accuracy, which we have summarized in Table 1. We provide full results, including additional models and perturbation bounds in Appendix A.9, and an evaluation of the considered highly accurate non-compositional models in Appendix A.10.

## 8 CONCLUSION

In this work, we address the robustness vs accuracy tradeoff by avoiding robust inaccuracy and leveraging model robustness as a selection mechanism. We present a new training method that jointly minimizes robust inaccuracy and maximizes robust accuracy. The key concept was extending an existing robust training loss with a term that minimizes robust inaccuracy, making our method widely applicable since it can be instantiated using various existing robust training methods. We show the practical benefits of our approach by both, using robustness as an abstain mechanism, and by leveraging compositional architectures to improve robustness without sacrificing accuracy.

However, there are also limitations and extensions to consider in future. First, while there are cases where our training improves robust accuracy and reduces robust inaccuracy, it does typically result in a trade-off between the two – reduced robust inaccuracy also leads to reduced robust accuracy. To address this issue, we compute a Pareto front of optimal solutions, all of which can be used to instantiate the compositional model. An interesting future work is exploring this trade-off further and develop new techniques to mitigate it. Second, given that we compute a Pareto front of optimal solutions, another extension is to consider model cascades that consist of different models along this Pareto front, and progressively fall back to models with higher robust accuracy but also higher robust inaccuracy. Third, we observed that the training becomes much harder as robust inaccuracy approaches zero (i.e., the best case). This is because these remaining robust inaccurate examples are the hardest to fix, and because there are only a few. In our work, we explored using data augmentation to address this issue, but more work is needed to make the training efficient in such a low data regime.

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

## A    APPENDIX

### A.1    DATASETS

We ran our evaluations on four different datasets, namely on `CIFAR-10` and `CIFAR-100` (Krizhevsky et al., 2009), the Mapillary Traffic Sign Dataset (`MTSD`) (Ertler et al., 2020), and a rail defect dataset provided by Swiss Federal Railways (`SBB`). Additionally, we used a synthetic dataset consisting of two-dimensional data points. In the following, we explain the necessary preprocessing steps to create the publicly available `MTSD` dataset.

**Mapillary Traffic Sign Dataset (MTSD)**    The Mapillary traffic sign dataset (Ertler et al., 2020) is a large-scale vision dataset that includes 52'000 fully annotated street-level images from all around the world. The dataset covers 400 known and other unknown traffic signs, resulting in over 255'000 traffic signs in total. Each street-level image is manually annotated and includes ground truth bounding boxes that locate each traffic sign in the image, as shown in Figure 5a. Further, each ground truth traffic sign annotation includes additional attributes such as ambiguousness or occlusion. Since the focus of this work is on classification, we convert the base `MTSD` dataset to a classification dataset (described below) by cropping to each ground truth bounding box. We show samples from the resulting cropped `MTSD` dataset in Figure 5b.

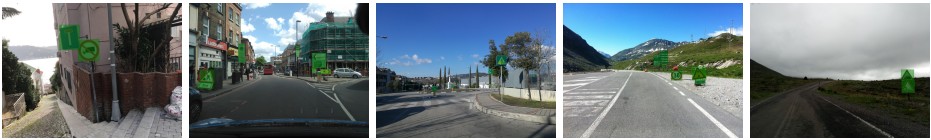

(a) Base Mapillary Traffic Sign Dataset (`MTSD`). The ground truth bounding boxes are visualized in green.

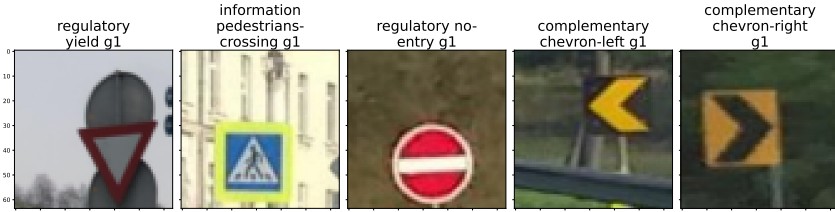

(b) Preprocessed Mapillary Traffic Sign Dataset (`MTSD`).

Figure 5: Illustration of Mapillary Traffic Sign Dataset (`MTSD`) samples. The base dataset consists of street-level images that include annotated ground truth bounding boxes locating the traffic signs (a). We convert the dataset to a classification task by cropping to the ground truth bounding boxes (b).

We convert the `MTSD` objection detection dataset into a classification dataset as follows:

1. Ignore all bounding boxes that are annotated as occluded (sign partly occluded), out-of-frame (sign cut off by image border), exterior (sign includes other signs), ambiguous (sign is not classifiable at all), included (sign is part of another bigger sign), dummy (looks like a sign but is not) (Ertler et al., 2020). Further, we ignore signs of class *other-sign*, since this is a general class that includes any traffic sign with a label not within the `MTSD` taxonomy.

2. Crop to all remaining bounding boxes and produce a labeled image classification dataset. Cropping is done with slack, i.e. we crop to a randomly upsized version of the original bounding box. Given a bounding box $BB = ([x_{min}, x_{max}], [y_{min}, y_{max}])$, the corresponding upsized bounding box is given as

$$UBB = \big([x_{min} - \lambda\alpha_x(x_{max} - x_{min}),\ x_{max} + \lambda(1 - \alpha_x)(x_{max} - x_{min})],$$
$$[y_{min} - \lambda\alpha_y(y_{max} - y_{min}),\ y_{max} + \lambda(1 - \alpha_y)(y_{max} - y_{min})]\big) \tag{16}$$

where $\alpha_x, \alpha_y \sim \mathcal{U}_{[0,1]}$ [3] and $\lambda$ is the slack parameter, which we set to $\lambda = 1.0$.

3. Resize cropped traffic signs to $(64, 64)$.

---

[3]$\mathcal{U}_{[a,b]}$ is the uniform distribution over the interval $[a, b]$.

**Rail Defect Dataset (SBB)**    The rail defect dataset (SBB) is a proprietary vision dataset collected and annotated by Swiss Federal Railways. It includes images of rails, each of which is annotated with ground truth bounding boxes for various types of rail defects. We note that all the models used in our work for this dataset are trained by the authors and not provided by SBB. In fact, for our work, we even consider a different type of task – classification instead of the original object detection. As a consequence, the accuracy and robustness results presented in our work are by no means representative of the actual models used by SBB.

## A.2    HYPERPARAMETERS

**TRADES**    We use $\mathcal{L}_{\text{TRADES}}$ (Zhang et al., 2019a) to both train models from scratch and fine-tune existing models. When training models from scratch, we train for 100 epochs using $\mathcal{L}_{\text{TRADES}}$, with an initial learning rate 1e-1, which we reduce to 1e-2 and 1e-3, once $75\%$ and $90\%$ of the total epochs are completed. When fine-tuning models, we train for 50 epochs using $\mathcal{L}_{\text{TRADES}}$, with an initial learning rate 1e-3, which we reduce to 1e-4 once $75\%$ of the total epochs are completed. We use batch size 200, use 10-step PGD (Madry et al., 2018) to generate adversarial examples during training, and set the $\beta$ parameter in $\mathcal{L}_{\text{TRADES}}$ to $\beta_{TRADES} = 6.0$.

**Empirical Robustness Abstain Training**    We fine-tune for 50 epochs using $\mathcal{L}_{\text{ERA}}$ (Equation 4), with an initial learning rate 1e-3, which we reduce to 1e-4 once $75\%$ of the total epochs are completed. We use batch size 200, use 10-step PGD (Madry et al., 2018) to generate adversarial examples during training, and set $\beta_{TRADES} = 6.0$ for the loss term $\mathcal{L}_{rob} = \mathcal{L}_{\text{TRADES}}$.

**MMA**    We use MMA (Ding et al., 2018) as an additional baseline to compare our models against. In our evaluations, we use the $d_{max} = {}^{12}/_{255}$ trained WideResNet-28-10 published by Ding et al. (2018), and fine-tune it using MMA with $d_{max} = {}^{4}/_{255}$ for 50 epochs. We decided to fine-tune with $d_{max} = {}^{4}/_{255}$, since we typically evaluate smaller perturbation regions ($\varepsilon_{\infty} \in \{{}^{1}/_{255}, {}^{2}/_{255}, {}^{4}/_{255}\}$), and since Ding et al. (2018) claim that $d_{max}$ should usually be set larger than $\varepsilon_{\infty}$ in standard adversarial training. We fine-tune for 50 epochs, using the same hyperparameters as Ding et al. (2018), and without using data augmentations.

**MART**    We use MART (Wang et al., 2019) as an additional baseline to compare our models against. In our evaluations, we use the $\varepsilon_{\infty} = {}^{8}/_{255}$ trained WideResNet-28-10 (trained with 500K unlabeled data) published by Wang et al. (2019), and fine-tune it using MART for the respective smaller perturbation region ($\varepsilon_{\infty} \in \{{}^{1}/_{255}, {}^{2}/_{255}, {}^{4}/_{255}\}$). We fine-tune for 50 epochs, using the same hyperparameters as Wang et al. (2019), and without using data augmentations.

**Certified Robustness Abstain Training**    We fine-tune for 50 epochs using $\mathcal{L}_{\text{CRA}}$ (Equation 7), with an initial learning rate 1e-3, which we reduce to 1e-4 once $75\%$ of the total epochs are completed. We use batch size 50, $k = 16$ *i.i.d.* samples from $\mathcal{N}(0, \sigma^2 \boldsymbol{I})$, and set the inverse softmax temperature to $\Gamma = 4.0$ (cf. Section 4.2).

**Probabilistic Certification via Randomized Smoothing**    We use the practical Monte Carlo algorithm by Cohen et al. (2019) for randomized smoothing, using the same certification hyperparameters as them. We use $N_0 = 100$ Monte Carlo samples to identify the most probable class $c_A$, $N = 100{,}000$ Monte Carlo samples to estimate a lower bound on the probability $p_A$, and set the failure probability to $\alpha = 0.001$.

**Synthetic Dataset**    In Figure 1, we illustrate the effect of our training on a synthetic three-class dataset, where each class follows a Gaussian distribution. We then use a simple four-layer neural network with 64 neurons per layer, and train it on $N = 1000$ synthetic samples, using $\mathcal{L}_{std}$, $\mathcal{L}_{\text{TRADES}}$ (Zhang et al., 2019a), and $\mathcal{L}_{\text{ERA}}$ (Equation 4). For each loss variant, we train for 20 epochs, use a fixed learning rate 1e-1, and batch size 10. For $\mathcal{L}_{\text{TRADES}}$ and $\mathcal{L}_{\text{ERA}}$, we use 10-step PGD (Madry et al., 2018) to generate adversarial examples during training, and set $\beta_{TRADES} = 6.0$.

Table 3: Robust accuracy ($\mathcal{R}_{rob}^{acc}$) and robust inaccuracy ($\mathcal{R}_{rob}^{\neg acc}$) of the $\mathcal{B}_{2/255}^{\infty}$ $\mathcal{L}_{\text{ERA}}$ ($\beta = 1.0$) finetuned Gowal et al. (2020) model, evaluated using both 40-step APGD (Croce & Hein, 2020b) and AutoAttack (Croce & Hein, 2020b).

| | $\mathcal{B}_{1/255}^{\infty}$ | | $\mathcal{B}_{2/255}^{\infty}$ | | $\mathcal{B}_{4/255}^{\infty}$ | | $\mathcal{B}_{8/255}^{\infty}$ | |
| | $\mathcal{R}_{rob}^{acc}$ | $\mathcal{R}_{rob}^{\neg acc}$ | $\mathcal{R}_{rob}^{acc}$ | $\mathcal{R}_{rob}^{\neg acc}$ | $\mathcal{R}_{rob}^{acc}$ | $\mathcal{R}_{rob}^{\neg acc}$ | $\mathcal{R}_{rob}^{acc}$ | $\mathcal{R}_{rob}^{\neg acc}$ |
|---|---|---|---|---|---|---|---|---|
| 40-step APGD | 92.93 | 1.01 | 86.45 | 0.33 | 64.09 | 0.06 | 17.20 | 0.0 |
| AutoAttack | 92.93 | 1.01 | 86.45 | 0.33 | 63.87 | 0.06 | 16.67 | 0.0 |

## A.3 ROBUSTNESS GUARANTEES FOR ROBUST SELECTION

Recall from Section 5 that, given an abstain model $(F_\theta, S)$ and a threat model $\mathcal{B}_\varepsilon^p(\boldsymbol{x}) := \{\boldsymbol{x}' : \|\boldsymbol{x}' - \boldsymbol{x}\|_p \leq \varepsilon\}$, $(F_\theta, S)$ is robustly selecting an input $\boldsymbol{x}$ if the selector $S$ selects all valid perturbations $\boldsymbol{x}' \in \mathcal{B}_\varepsilon^p(\boldsymbol{x})$:

$$\mathcal{R}_{rob}^{sel}(S) = \mathbb{E}_{(\boldsymbol{x},y) \sim \mathcal{D}} \quad \mathbf{1}\{\forall \boldsymbol{x}' \in \mathcal{B}_\varepsilon^p(\boldsymbol{x}). \, S(\boldsymbol{x}') = 1\}$$

Further, recall that when evaluating the robustness of an empirical robustness indicator selector $S_{\text{ERI}}$ (Equation 10), we in fact need to check robustness of the model $F_\theta$ to double the perturbation region $\boldsymbol{x}' \in \mathcal{B}_{2 \cdot \varepsilon}^p(\boldsymbol{x})$, which can be see from the following derivation:

$$\begin{aligned}
\mathcal{R}_{rob}^{sel}(S_{\text{ERI}}) &= \mathbb{E}_{(\boldsymbol{x},y) \sim \mathcal{D}} \quad \mathbf{1}\{\forall \boldsymbol{x}' \in \mathcal{B}_\varepsilon^p(\boldsymbol{x}). \, S_{\text{ERI}}(\boldsymbol{x}') = 1\} \\
&= \mathbb{E}_{(\boldsymbol{x},y) \sim \mathcal{D}} \quad \mathbf{1}\{\forall \boldsymbol{x}' \in \mathcal{B}_\varepsilon^p(\boldsymbol{x}). \, \mathbf{1}\{\forall \boldsymbol{x}'' \in \mathcal{B}_\varepsilon^p(\boldsymbol{x}'). \, F_\theta(\boldsymbol{x}'') = F_\theta(\boldsymbol{x}')\}\} \\
&= \mathbb{E}_{(\boldsymbol{x},y) \sim \mathcal{D}} \quad \mathbf{1}\{\forall \boldsymbol{x}' \in \mathcal{B}_{2 \cdot \varepsilon}^p(\boldsymbol{x}). \, F_\theta(\boldsymbol{x}') = F_\theta(\boldsymbol{x})\}
\end{aligned}$$

## A.4 COMPARING APGD AND AUTOATTACK ROBUSTNESS

Recall from Section 7 that we use 40-step $\text{APGD}_{\text{CE}}$ (Croce & Hein, 2020b) (referred to as APGD) to evaluate the empirical robustness of classifiers $F_\theta$. APGD is one of the adversarial attacks that constitute AutoAttack (Croce & Hein, 2020b), which is an ensemble of adversarial attacks. Concretely, AutoAttack consists of $\text{APGD}_{\text{CE}}$ (Croce & Hein, 2020b), $\text{APGD}^{\text{T}}_{\text{DLR}}$ (Croce & Hein, 2020b), $\text{FAB}^{\text{T}}$ (Croce & Hein, 2020a), and SquareAttack (Andriushchenko et al., 2020).

In the following, we conduct an ablation study over 40-step APGD and AutoAttack by comparing the robustness of an $\mathcal{L}_{\text{ERA}}$ trained model. Concretely, we consider the Gowal et al. (2020) WideResNet-28-10 model, which was finetuned for $\mathcal{B}_{2/255}^{\infty}$ using our $\mathcal{L}_{\text{ERA}}$ loss (with $\beta = 1.0$) on CIFAR-10(cf. Section 7.1). We then evaluate its robust accuracy $\mathcal{R}_{rob}^{acc}$ and robust inaccuracy $\mathcal{R}_{rob}^{\neg acc}$ for the threat models $\varepsilon_\infty \in \{1/255, 2/255, 4/255, 8/255\}$, using both 40-step APGD and AutoAttack, and show the results in Table 3. Observe that for small perturbation regions $\varepsilon_\infty \in \{1/255, 2/255\}$, the robust accuracy and robust inaccuracy are equivalent for 40-step APGD and AutoAttack, whereas for larger perturbation regions $\varepsilon_\infty \in \{4/255, 8/255\}$, AutoAttack robust accuracy is marginally lower than 40-step APGD robust accuracy.

## A.5 COMPARING ADVERSARIES FOR SOFTMAX RESPONSE (SR)

Recall from Section 7.2 that we evaluated the robustness of softmax response (SR) abstain models using APGDconf, which is a modified version of APGD (Croce & Hein, 2020b) using the alternative adversarial attack objective by Stutz et al. (2020). This modified objective optimizes for an adversarial example $\boldsymbol{x}'$ that maximizes the confidence in any label $c \neq F_\theta(\boldsymbol{x})$, instead of minimizing the confidence in the predicted label:

$$\boldsymbol{x}' = \arg \max_{\hat{\boldsymbol{x}} \in \mathcal{B}_\varepsilon^p(\boldsymbol{x})} \max_{c \neq F_\theta(\boldsymbol{x})} f_\theta(\hat{\boldsymbol{x}})_c \tag{17}$$

The resulting adversarial attack finds high confidence adversarial examples, and thus represents an effective attack against a softmax response selector $S_{\text{SR}}$.

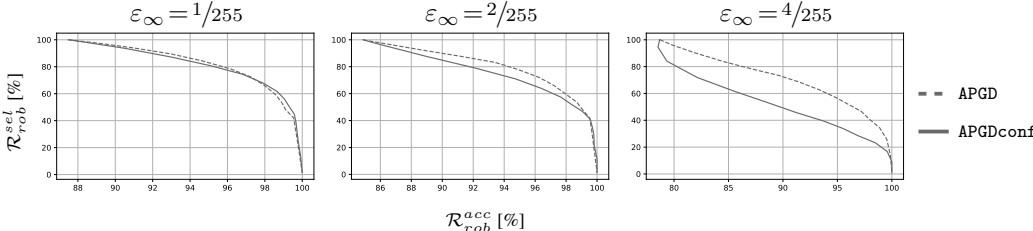

Figure 6: Robust selection ($\mathcal{R}_{rob}^{sel}$) and robust accuracy ($\mathcal{R}_{rob}^{acc}$) for `CIFAR-10` softmax response (SR) abstain models $(F, S_{\text{SR}})$, for varying threshold $\tau \in [0, 1)$ and using the WideResNet-28-10 classifier $F$ by Carmon et al. (2019). Each SR abstain model is evaluated via `APGD` (Croce & Hein, 2020b) and `APGDconf` (Equation 17).

Table 4: Robust selection ($\mathcal{R}_{rob}^{sel}$) and robust accuracy ($\mathcal{R}_{rob}^{acc}$) of empirical robustness indicator abstain models $(F, S_{\text{ERI}})$, trained using $\mathcal{L}_{\text{ERA}}$ (Equation 4) and $\mathcal{L}_{\text{DGA}}$ (Equation 18).

| CIFAR-10 | | $\mathcal{B}_{1/255}^{\infty}$ | | $\mathcal{B}_{2/255}^{\infty}$ | |
| --- | --- | --- | --- | --- | --- |
| Pre-trained Model | Finetuning | $\mathcal{R}_{rob}^{sel}$ | $\mathcal{R}_{rob}^{acc}$ | $\mathcal{R}_{rob}^{sel}$ | $\mathcal{R}_{rob}^{acc}$ |
| | $\mathcal{L}_{\text{ERA}}$ | **86.31** | **96.63** | **78.24** | **97.33** |
| Zhang et al. (2019a) | $\mathcal{L}_{\text{DGA}}$ | 84.98 | 94.92 | 75.73 | 96.22 |
| (ResNet-50) | $\mathcal{L}_{\text{ERA}} + AA$ | **83.44** | **97.47** | **74.63** | **98.31** |
| | $\mathcal{L}_{\text{DGA}} + AA$ | 80.72 | 96.56 | 73.59 | 97.88 |

In the following, we conduct an ablation study over `APGD` and `APGDconf` by evaluating the robust selection $\mathcal{R}_{rob}^{sel}$ and robust accuracy $\mathcal{R}_{rob}^{acc}$ of an SR abstain model $(F_\theta, S_{\text{SR}})$ using both `APGD` and `APGDconf`. We use the adversarially trained WideResNet-28-10 model by Carmon et al. (2019) (taken from RobustBench (Croce et al., 2020)), trained on `CIFAR-10` for $\varepsilon_\infty = 8/255$ perturbations. We then evaluate the classifier as an SR abstain model $(F_\theta, S_{\text{SR}})$ with varying threshold $\tau \in [0, 1)$, and report the robust selection and robust accuracy for varying $\ell_\infty$ perturbations in Figure 6. Observe that for small perturbations such as $\varepsilon_\infty = 1/255$, `APGD` and `APGDconf` are mostly equivalent concerning robust selection and robust accuracy. However, for larger perturbations such as $\varepsilon_\infty = 4/255$, the SR abstain model is significantly less robust to `APGDconf` than to standard `APGD`, showing the importance of choosing a suitable adversarial attack. High confidence adversarial examples are generally more likely to be found for larger perturbations, thus an SR selector is significantly less robust to `APGDconf` than to `APGD` for larger perturbations.

### A.6 LOSS FUNCTION ABLATION STUDY

Additionally to the $\mathcal{L}_{\text{ERA}}$ loss from Equation 4, we consider an alternative loss formulation for training an empirical robustness indicator abstain model. The formulation is based on the Deep Gamblers loss (Liu et al., 2019), which considers an abstain model $(F_\theta, S)$ with an explicit abstain class $a$ as a selection mechanism. Since we consider robustness indicator selection, we replace the output probability of the abstain class $f_\theta(\boldsymbol{x})_a$ with the output probability of the most likely adversarial label. This corresponds to the probability of a sample being non-robust and thus the probability of abstaining under a robustness indicator selector. Similar to $\mathcal{L}_{\text{ERA}}$, we also add the `TRADES` loss (Zhang et al., 2019a) to optimize robust accuracy. The resulting loss is then defined as:

$$\mathcal{L}_{\text{DGA}}(f_\theta, (\boldsymbol{x}, y)) = \beta \cdot \mathcal{L}_{\text{TRADES}}(f_\theta, (\boldsymbol{x}, y)) - \log\left(f_\theta(\boldsymbol{x})_y + \max_{c \in \mathcal{Y} \setminus \{F_\theta(\boldsymbol{x})\}} f_\theta(\boldsymbol{x}')_c\right) \quad (18)$$

We conduct an ablation study over the two loss functions, $\mathcal{L}_{\text{ERA}}$ and $\mathcal{L}_{\text{DGA}}$, for `CIFAR-10` and a $\varepsilon_\infty = 8/255$ `TRADES` (Zhang et al., 2019a) trained ResNet-50 model. We fine-tune the model for $\ell_\infty$ perturbations of radii $1/255$ and $2/255$, using both $\mathcal{L}_{\text{ERA}}$ and $\mathcal{L}_{\text{DGA}}$, training for 50 epochs each and setting the regularization parameter $\beta = 1.0$. For each loss variant, we train the base model once without data augmentations and once using the AutoAugment (AA) policy (Cubuk et al., 2018).

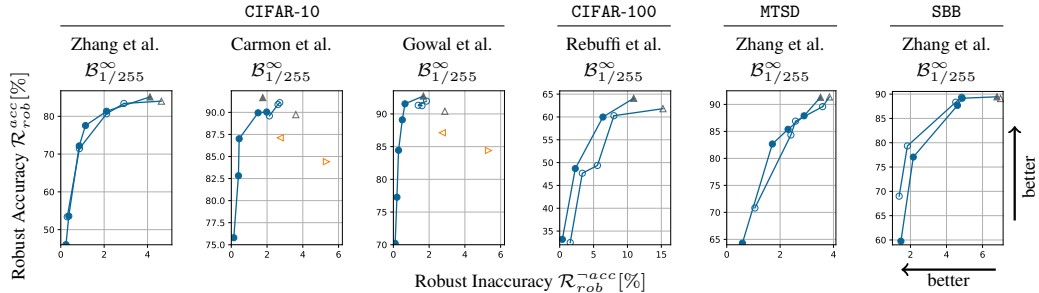

Figure 7: Robust accuracy ($\mathcal{R}_{rob}^{acc}$) and robust inaccuracy ($\mathcal{R}_{rob}^{\neg acc}$) of existing robust models ($\triangle$, $\blacktriangle$) fine-tuned with our proposed loss ($\bigcirc$, $\bullet$). Further, we also show models finetuned via MART (Wang et al., 2019) ($\triangleleft$) and MMA (Ding et al., 2018) ($\triangleright$). Our approach consistently reduces the number of robust inaccurate samples across various datasets, existing models and at different regularization levels $\beta$.

We show the robust accuracy and the robust selection of the resulting robustness indicator abstain models in Table 4. Observe that for all experiments, $\mathcal{L}_{\text{ERA}}$ trained models achieve consistently higher robust accuracy and higher robust selection, compared to $\mathcal{L}_{\text{DGA}}$ trained models. For instance, when training for $\varepsilon_\infty = 1/255$ perturbations without data augmentations, $\mathcal{L}_{\text{ERA}}$ achieves $+1.71\%$ higher robust accuracy and $+1.33\%$ higher robust selection, compared to $\mathcal{L}_{\text{DGA}}$. Similarly, when training with AutoAugment, $\mathcal{L}_{\text{ERA}}$ achieves $+0.91\%$ higher robust accuracy and $+2.72\%$ higher robust selection. Similar results hold for $\varepsilon_\infty = 2/255$ perturbations.

### A.7 ADDITIONAL EXPERIMENTS ON REDUCING ROBUST INACCURACY

In this section, we present additional experiments on reducing robust inaccuracy for empirical robustness.

Similar to the results in Figure 2, we show the robust accuracy ($\mathcal{R}_{rob}^{acc}$) and robust inaccuracy ($\mathcal{R}_{rob}^{\neg acc}$) of different existing models fine-tuned with ($\blacktriangle$) and without ($\triangle$) data augmentations, in Figure 7. At the same time, Figure 7 also shows the same models fine-tuned with our proposed loss with ($\bullet$) and without ($\bigcirc$) data augmentations. We again observe that our approach achieves consistently lower robust robust inaccuracy, compared to existing robust models. For example, on CIFAR-10 and for $\mathcal{B}_{1/255}^\infty$, the model from Carmon et al. (2019) achieves $91.7\%$ robust accuracy but also $1.8\%$ robust inaccuracy. Using our loss $\mathcal{L}_{\text{ERA}}$ and varying the regularization term $\beta$, we can obtain a number of models that reduce robust inaccuracy to $0.14\%$ while still achieving robust accuracy of $75.8\%$.

### A.8 ADDITIONAL EXPERIMENTS ON USING ROBUSTNESS TO ABSTAIN

In this section, we present additional experiments on comparing different abstain approaches for empirical robustness.

We compare robustness indicator abstain models ($F, S_{\text{RI}}$) using existing robust classifiers $\text{TRADES}_{\text{RI}}$ and classifiers fine-tuned with our proposed loss $\text{ERA}_{\text{RI}}$. Further, we again consider softmax response and selection network abstain models, as described in Section 7.2. Equivalent to Section 7.2, we use the robust selection ($\mathcal{R}_{rob}^{sel}$), and the ratio of non-abstained samples that are robust and accurate ($\mathcal{R}_{rob}^{acc}$) as our evaluation metrics.

We show the comparison of the different abstain models in Figure 8. Similar to the results in Section 7.2, we again show that, as designed, our approach consistently improves robust accuracy. For instance, consider the CIFAR-10 Zhang et al. (2019a) model at $\varepsilon_\infty = 1/255$, trained without data augmentations ($\bigcirc$). The $\text{ERA}_{\text{RI}}$ model with the highest robust selection $\mathcal{R}_{rob}^{sel}$ improves robust accuracy by $+2.39\%$ at the expense of $-3.44\%$ decrease in robust selection. This tradeoff is close to optimal since our approach increases robust accuracy by correctly abstaining from mispredicted samples, thus an increase in robust accuracy results in a corresponding decrease in robust selection. Further, we again observe that by varying the regularization parameter $\beta$, we can obtain a Pareto front of optimal solutions. Considering the CIFAR-10 Zhang et al. (2019a) model at $\varepsilon_\infty = 1/255$, trained with data augmentations ($\bullet$), we can improve the robust accuracy up to $99.75\%$, an increase

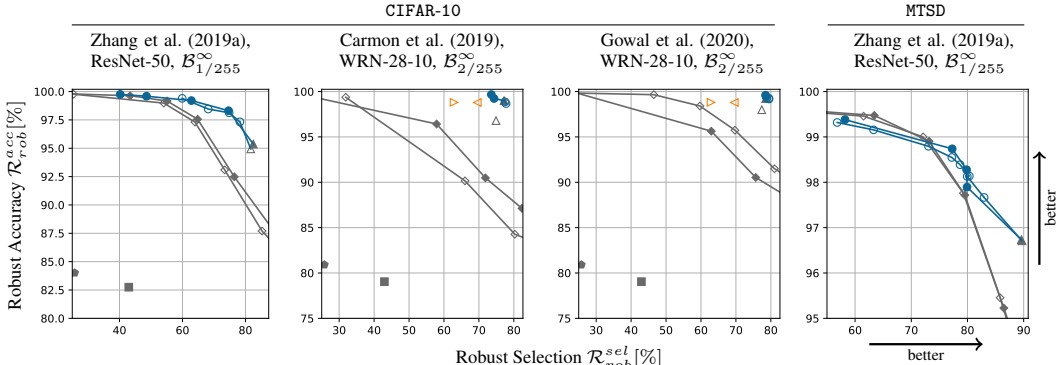

Figure 8: Comparison of different abstain approaches including existing robust classifiers TRADES$_{\text{RI}}$ (△, ▲), MART$_{\text{RI}}$ (◀), MMA$_{\text{RI}}$ (▷), classifiers fine-tuned with our proposed loss ERA$_{\text{RI}}$ (○, ●), selection network (■, ◖), and softmax response (◇, ◆) abstain models. The higher $\mathcal{R}_{rob}^{sel}$ and $\mathcal{R}_{rob}^{acc}$, the better (top right corner is optimal).

of +4.38% compared to the corresponding TRADES$_{\text{RI}}$ model (▲). However, this comes at the expense of a disproportionally large decrease of -42.27% lower robust selection. We observe similar results for other models, datasets, and perturbations regions, shown in Figure 8.

Further, we again note that our approach mostly improves both robust selection and robust accuracy when compared to softmax response and selection network abstain models.

## A.9 ADDITIONAL EXPERIMENTS ON BOOSTING ROBUSTNESS WITHOUT ACCURACY LOSS

In this section, we present additional results on combining abstain models with state-of-the-art models trained to achieve high natural accuracy.

Equivalent to Section 7.3, we put the abstain models trained so far in 2-composition (Section 6) with the standard trained core models discussed in Appendix A.10. We show the natural ($\mathcal{R}_{nat}$) and adversarial accuracy ($\mathcal{R}_{rob}^{acc}$) of the resulting 2-compositional architectures in Figure 9.

We again observe that 2-compositional architectures using models trained by our method (○, ●) improve over existing methods that solely optimize for robust accuracy (△, ▲). Further, our method mostly improves both the natural and robust accuracy, compared to 2-compositional architectures using softmax response (◇, ◆) or selection network (■, ◖) to abstain. For example, on SBB and the Zhang et al. (2019a) model at $\varepsilon_{\infty} = 1/255$, our approach (○) improves natural accuracy by +0.68%, while decreasing the robust accuracy by only -1.54%.

Further, we show that 2-compositional architectures using models trained by our method achieve significantly higher robustness and mostly equivalent overall accuracy, compared to state-of-the-art non-compositional models trained for high natural accuracy. In Table 5, we show the natural ($\mathcal{R}_{nat}$) and adversarial accuracy ($\mathcal{R}_{rob}^{acc}$) of our 2-compositional models and illustrate the accuracy improvement over the standard trained models discussed in Appendix A.10. For instance, consider CIFAR-10 at $\varepsilon_{\infty} = 2/255$ and the 2-compositional architecture using the Gowal et al. (2020) model as robust model $F_{robust}$. Our model improves the robust accuracy by +75.3% and the natural accuracy by +0.1%, compared to the standard trained model by Zhao et al. (2020). Similar results hold for other models, datasets, and perturbation regions.

## A.10 CORE MODELS

Recall from Section 6 that an abstain model $(F, S)$ can be enhanced by a core model $F_{core}$, which makes a prediction on all abstained samples, resulting in 2-compositional architectures. In Section 7.3, we presented an evaluation of 2-compositional architectures, where we used state-of-the-art standard trained models as core models. In Table 6, we show the natural and adversarial accuracy of core

Table 5: Improvements of 2-compositional architectures using models $F_{robust}$ trained with our method over non-compositional models trained to optimize natural accuracy only (Appendix A.10).

| | | CIFAR-10 | | CIFAR-100 | MTSD | SBB |
|---|---|---|---|---|---|---|
| | $F_{core}$ | Zhao et al. (2020) | | (WideResNet-28-10) | (ResNet-50) | (ResNet-50) |
| | $F_{robust}$ | Carmon et al. | Gowal et al. | Rebuffi et al. | Zhang et al. | Zhang et al. |
| $\mathcal{B}^{\infty}_{1/255}$ | $\mathcal{R}^{acc}_{rob}$ | 86.5 $^{(+60.3\%)}$ | 87.8 $^{(+61.6\%)}$ | 44.0 $^{(+24.1\%)}$ | 84.5 $^{(+9.8\%)}$ | 88.4 $^{(+12.7\%)}$ |
| | $\mathcal{R}_{nat}$ | 97.6 $^{(-0.2\%)}$ | 98.0 $^{(+0.2\%)}$ | 80.5 $^{(+0.3\%)}$ | 94.1 $^{(+0.3\%)}$ | 92.3 $^{(+0.9\%)}$ |
| $\mathcal{B}^{\infty}_{2/255}$ | $\mathcal{R}^{acc}_{rob}$ | 73.4 $^{(+70.5\%)}$ | 78.2 $^{(+75.3\%)}$ | 41.9 $^{(+38.8\%)}$ | 69.9 $^{(+29.2\%)}$ | 82.4 $^{(+37.7\%)}$ |
| | $\mathcal{R}_{nat}$ | 97.8 $^{(+0.0\%)}$ | 97.9 $^{(+0.1\%)}$ | 80.18 $^{(+0.01\%)}$ | 94.0 $^{(+0.2\%)}$ | 91.3 $^{(-0.1\%)}$ |

models used in Section 7.3, for varying $\ell_\infty$ perturbation regions, where we use 40-step APGD (Croce & Hein, 2020b) to evaluate robustness.

Table 6: Natural ($\mathcal{R}_{nat}$) and adversarial accuracy ($\mathcal{R}^{acc}_{rob}$) of standard trained core models, used in 2-compositional architectures in Section 7.3 and Appendix A.9.

| Dataset | Model $F_{core}$ | $\mathcal{R}_{nat}$ [%] | $\mathcal{R}^{acc}_{rob}$ [%] | | |
|---|---|---|---|---|---|
| | | | $\mathcal{B}^{\infty}_{1/255}$ | $\mathcal{B}^{\infty}_{2/255}$ | $\mathcal{B}^{\infty}_{4/255}$ |
| CIFAR-10 | Zhao et al. (2020) (WideResNet-40-10) | 97.81 | 26.18 | 2.92 | 0.06 |
| CIFAR-100 | (WideResNet-28-10) | 80.17 | 19.9 | 3.06 | 0.15 |
| MTSD | (ResNet-50) | 93.79 | 74.66 | 40.71 | 7.51 |
| SBB | (ResNet-50) | 91.37 | 75.65 | 44.69 | 8.76 |

## A.11 ROBUSTNESS/ACCURACY DATASET SPLITS

Consider a robustness indicator abstain model $(F_\theta, S_{\mathrm{RI}})$ and a labeled dataset $D = \{(\boldsymbol{x}_i, y_i)_{i=1}^N\}$ on which we evaluate the classifier $F_\theta \colon \mathcal{X} \to \mathcal{Y}$. Based on the robustness and accuracy of the classifier $F_\theta$, we can partition $D$ into four disjoint subsets $D = \{D_{F_\theta}^{r \wedge a}, D_{F_\theta}^{\neg r \wedge a}, D_{F_\theta}^{r \wedge \neg a}, D_{F_\theta}^{\neg r \wedge \neg a}\}$, where:

$$D_{F_\theta}^{r \wedge a} = \{(\boldsymbol{x}, y) \in D \colon \forall \boldsymbol{x}' \in \mathcal{B}_\varepsilon^p(\boldsymbol{x}). \, F_\theta(\boldsymbol{x}') = F_\theta(\boldsymbol{x}) \wedge F_\theta(\boldsymbol{x}) = y\}$$

$$D_{F_\theta}^{r \wedge \neg a} = \{(\boldsymbol{x}, y) \in D \colon \forall \boldsymbol{x}' \in \mathcal{B}_\varepsilon^p(\boldsymbol{x}). \, F_\theta(\boldsymbol{x}') = F_\theta(\boldsymbol{x}) \wedge F_\theta(\boldsymbol{x}) \neq y\}$$

$$D_{F_\theta}^{\neg r \wedge a} = \{(\boldsymbol{x}, y) \in D \colon \exists \boldsymbol{x}' \in \mathcal{B}_\varepsilon^p(\boldsymbol{x}). \, F_\theta(\boldsymbol{x}') \neq F_\theta(\boldsymbol{x}) \wedge F_\theta(\boldsymbol{x}) = y\}$$

$$D_{F_\theta}^{\neg r \wedge \neg a} = \{(\boldsymbol{x}, y) \in D \colon \exists \boldsymbol{x}' \in \mathcal{B}_\varepsilon^p(\boldsymbol{x}). \, F_\theta(\boldsymbol{x}') \neq F_\theta(\boldsymbol{x}) \wedge F_\theta(\boldsymbol{x}) \neq y\}$$

We illustrate this dataset partitioning on the CIFAR-10 (Krizhevsky et al., 2009) dataset. We consider a TRADES (Zhang et al., 2019b) trained ResNet-50 and the WideResNet-28-10 models by Carmon et al. (2019); Gowal et al. (2020) (taken from Robustbench (Croce et al., 2020)), where each model is adversarially pretrained for $\varepsilon_\infty = 8/255$ and then fine-tuned via TRADES to the respective $\ell_\infty$ threat model illustrated Table 7. Further, we also consider a standard trained ResNet-50. We then evaluate the robustness and accuracy of each model using 40-step APGD (Croce & Hein, 2020b). Considering Table 7, note that standard adversarial training methods do not necessarily eliminate the occurrence of robust inaccurate samples $(\boldsymbol{x}, y) \in D_{F_\theta}^{r \wedge \neg a}$, and that the robust inaccuracy generally increases for smaller perturbation regions. Further, we note that while standard trained models have low robust inaccuracy, they also have low overall robustness, resulting in low overall robust accuracy.

Further, we also illustrate the robustness-accuracy dataset partitioning on CIFAR-100 (Krizhevsky et al., 2009). We consider a standard trained WideResNet-28-10 and the adversarially trained WideResNet-28-10 by Rebuffi et al. (2021). Again, the model by Rebuffi et al. (2021) was pretrained for $\varepsilon_\infty = 8/255$ perturbations and then TRADES fine-tuned for the respective threat model indicated in Table 8. We again evaluate the robustness-accuracy dataset partitioning for varying $\ell_\infty$ perturbations using 40-step APGD (Croce & Hein, 2020b), and list the exact size of each data split in Table 8.

Notably, we observe that on the model by Rebuffi et al. (2021), $15.24\%$ of all test samples are robust but inaccurate for $\varepsilon_\infty = 1/255$ perturbations, which is a significantly larger fraction compared to similar models on CIFAR-10.

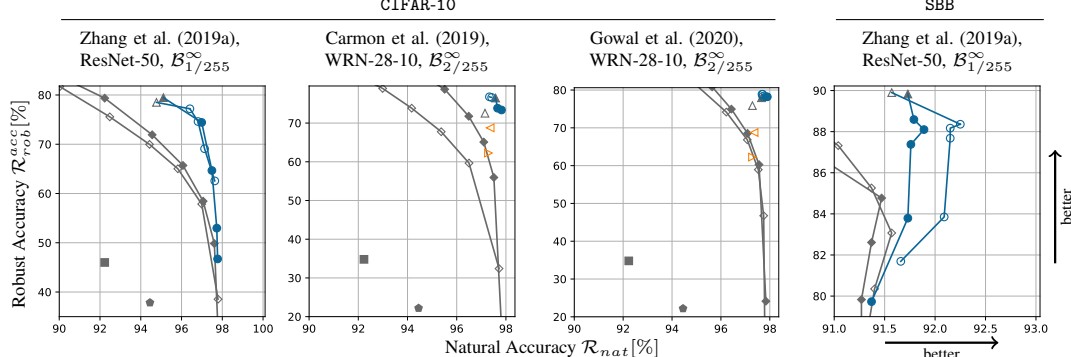

Figure 9: Natural ($\mathcal{R}_{nat}$) and robust accuracy ($\mathcal{R}_{rob}^{acc}$) for 2-compositional ERA$_{RI}$ models (○, ●), and 2-compositional TRADES$_{RI}$ (△, ▲), MART$_{RI}$ (◁), and MMA$_{RI}$ (▷) models. Further, we also consider 2-compositional ACE-COLT$_{SN}$, ACE-IBP$_{SN}$ (■, ⬠), and 2-compositional TRADES$_{SR}$ (◇, ◆) models. The core models used in the compositional architectures are listed in Appendix A.10.

Table 7: CIFAR-10 robustness-accuracy dataset partitioning. We consider a TRADES (Zhang et al., 2019a) trained ResNet-50, adversarially trained WideResNet-28-10 models (Carmon et al., 2019; Gowal et al., 2020), and a standard trained ResNet-50. Adversarially trained models are trained for the respective perturbation region. Each model is evaluated for the indicated $\ell_\infty$ threat model, using 40-step APGD (Croce & Hein, 2020b).

| Threat Model | Data Split | Relative Split Size [%] | | | |
|---|---|---|---|---|---|
| | | Zhang et al. (ResNet-50) | Carmon et al. (WRN-28-10) | Gowal et al. (WRN-28-10) | $\mathcal{L}_{std}$ (ResNet-50) |
| $\mathcal{B}_{1/255}^\infty$ | ■$\lvert D_{F_\theta}^{\neg r \wedge \neg a}\rvert$ | 5.17 | 3.33 | 2.85 | 6.97 |
| | ■$\lvert D_{F_\theta}^{r \wedge \neg a}\rvert$ | 4.64 | 3.61 | 2.88 | 0.0 |
| | ■$\lvert D_{F_\theta}^{\neg r \wedge a}\rvert$ | 6.18 | 3.32 | 3.87 | 74.89 |
| | ■$\lvert D_{F_\theta}^{r \wedge a}\rvert$ | 84.01 | 89.74 | 90.40 | 18.14 |
| $\mathcal{B}_{2/255}^\infty$ | ■$\lvert D_{F_\theta}^{\neg r \wedge \neg a}\rvert$ | 7.94 | 7.38 | 4.86 | 6.97 |
| | ■$\lvert D_{F_\theta}^{r \wedge \neg a}\rvert$ | 4.13 | 2.40 | 2.25 | 0.0 |
| | ■$\lvert D_{F_\theta}^{\neg r \wedge a}\rvert$ | 10.38 | 3.20 | 6.74 | 91.80 |
| | ■$\lvert D_{F_\theta}^{r \wedge a}\rvert$ | 77.55 | 87.02 | 86.15 | 1.23 |
| $\mathcal{B}_{4/255}^\infty$ | ■$\lvert D_{F_\theta}^{\neg r \wedge \neg a}\rvert$ | 13.42 | 8.23 | 6.64 | 6.97 |
| | ■$\lvert D_{F_\theta}^{r \wedge \neg a}\rvert$ | 3.31 | 1.05 | 0.87 | 0.0 |
| | ■$\lvert D_{F_\theta}^{\neg r \wedge a}\rvert$ | 17.19 | 16.87 | 15.96 | 93.03 |
| | ■$\lvert D_{F_\theta}^{r \wedge a}\rvert$ | 66.08 | 73.85 | 76.53 | 0.0 |
| $\mathcal{B}_{8/255}^\infty$ | ■$\lvert D_{F_\theta}^{\neg r \wedge \neg a}\rvert$ | 18.17 | 9.55 | 9.21 | 6.97 |
| | ■$\lvert D_{F_\theta}^{r \wedge \neg a}\rvert$ | 2.64 | 0.76 | 1.31 | 0.0 |
| | ■$\lvert D_{F_\theta}^{\neg r \wedge a}\rvert$ | 29.79 | 27.82 | 23.78 | 93.03 |
| | ■$\lvert D_{F_\theta}^{r \wedge a}\rvert$ | 49.40 | 61.87 | 65.70 | 0.0 |

Table 8: CIFAR-100 robustness-accuracy dataset partitioning. We consider a standard trained WideResNet-28-10 and the adversarially trained WideResNet-28-10 by Rebuffi et al. (2021), trained for the respective perturbation region considered in each evaluation. Each model is evaluated for the indicated $\ell_\infty$ threat model, using 40-step APGD (Croce & Hein, 2020b).

| Threat Model | Data Split | Relative Split Size [%] | |
|---|---|---|---|
| | | Rebuffi et al. (WRN-28-10) | $\mathcal{L}_{std}$ (WRN-28-10) |
| $\mathcal{B}^\infty_{1/255}$ | $\|D_{F_\theta}^{\neg r \wedge \neg a}\|$ | 15.20 | 19.80 |
| | $\|D_{F_\theta}^{r \wedge \neg a}\|$ | 15.24 | 0.03 |
| | $\|D_{F_\theta}^{\neg r \wedge a}\|$ | 7.75 | 60.27 |
| | $\|D_{F_\theta}^{r \wedge a}\|$ | 61.81 | 19.9 |
| $\mathcal{B}^\infty_{2/255}$ | $\|D_{F_\theta}^{\neg r \wedge \neg a}\|$ | 32.75 | 19.82 |
| | $\|D_{F_\theta}^{r \wedge \neg a}\|$ | 8.71 | 0.01 |
| | $\|D_{F_\theta}^{\neg r \wedge a}\|$ | 5.11 | 77.11 |
| | $\|D_{F_\theta}^{r \wedge a}\|$ | 53.43 | 3.06 |
| $\mathcal{B}^\infty_{4/255}$ | $\|D_{F_\theta}^{\neg r \wedge \neg a}\|$ | 30.57 | 19.83 |
| | $\|D_{F_\theta}^{r \wedge \neg a}\|$ | 4.34 | 0.0 |
| | $\|D_{F_\theta}^{\neg r \wedge a}\|$ | 23.16 | 80.02 |
| | $\|D_{F_\theta}^{r \wedge a}\|$ | 41.93 | 0.15 |
| $\mathcal{B}^\infty_{8/255}$ | $\|D_{F_\theta}^{\neg r \wedge \neg a}\|$ | 33.70 | 19.83 |
| | $\|D_{F_\theta}^{r \wedge \neg a}\|$ | 3.91 | 0.0 |
| | $\|D_{F_\theta}^{\neg r \wedge a}\|$ | 26.66 | 80.17 |
| | $\|D_{F_\theta}^{r \wedge a}\|$ | 35.73 | 0.0 |

