# OpenReview forum: "Just Avoid Robust Inaccuracy: Boosting Robustness Without Sacrificing Accuracy"
_ICLR.cc/2023/Conference — Submitted to ICLR 2023_

### Official Review · Reviewer_KM7o · 2022-10-24

**Confidence:** 3
**Correctness:** 2
**Technical Novelty And Significance:** 3
**Empirical Novelty And Significance:** 3
**Recommendation:** 3

**Clarity, Quality, Novelty And Reproducibility:**

- **Clarity**: Although the structure of the paper is very clean and it has a nice direct tone, the main empirical section of this work is very hard to read and confusing.
- **Quality**: Because all results are provided for non-standard and small $\epsilon$ values, I have doubts that the findings of this work are completely credible.
- **Novelty**: As far as I know, this work is novel.
- **Reproducible**: All experimental details are thoroughly discussed in the Appendix.

**Strength And Weaknesses:**

# Strengths

1. **Interesting propoed method**: The abstain mechanism proposed in this work is an interesting concept with nice connections to the confidence calibration literature. Personally, I have found the idea of calibrating the robustness of a neural network a very appealing concept and an interesting research line.
2. **Direct writing style**: I really appreciate the direct tone of the paper that goes directly to the main ideas after a short motivation. The paper structure is also very clean.

# Weaknesses

1. **Very unclear evaluation objective and presentation of results**: The main weakness of this work is that the main results of the experiments are presented in a very convoluted fashion. Despite several attempts at carefully reading the results Section 7 and inspecting the provided plots, I still am not sure which setting in the plots corresponds to the discussed numbers in the text, when can we say that a method is superior to another, or how are the different evaluation metrics measured in the context of the abstain models. Let me give a few examples of my confusion:
   - Several times during the paper, the authors argue that their proposed method can strik a better balance between robust and natural accuracy. In fact, they highlight certain instances of this in the text. However, in all plots I always observe that the TRADES models lie exactly on the same Pareto frontier as the ones trained with the proposed protocols. If this is the case, then what is the added value of the proposed methods? Reaching different points in the Pareto front? If so, did the authors evaluate make sure that tuning the regularization strength of TRADES does not cover the full Pareto front as well?
   - In page 3, the robust accuracy is defined using the standard notion, i.e., the percentage of points that are robust and accurate in the test set. However, later in page 8, the robust accuracy is defined as the percentage of points that are robust, accurate AND for which the model did not abstain. This is a big difference, as one could artificially increase robustness, by increasing the number of points for which the model abstains.
2. **Robustness evaluations only at very small $\epsilon$ values**: I find very surprising that all the results of this paper are given for non-standard $\epsilon$ values. Especially, when the pretrained models used by the authors to finetune using their new loss where trained using larger $\epsilon$ values. This makes me very suspicious that many of the results provided in this work might not hold for larger $\epsilon$. If this was the case, this would seriously decrease the credibility of the findings and make the strong statements of the abstract and introduction much weaker.
3. (Minor) **Hard-to-read plots**: I personally find the use of markers with and without filling, and different colors, to denote different methods with different training protocols very confusing and hard-to-read.




**Summary Of The Paper:**

This paper proposes a novel approach to improve the tradeoff between adversarial robustness and natural accuracy based on an abstain mechanism that discards datapoints which are not robust to its own predictions. To boos the performance of such model the authors propose a training protocol which minimizes the percentage of training points which are robust by innacurate. The paper claims that this training protocol can be applied only in a second fine-tuning stage still yielding good results. Results are provided comparing the proposed approach against different robustness benchmarks. The authors argue that their models consistently strike a better tradeoff between robustness and accuracy, while also minimizing robust innacuracy.

**Summary Of The Review:**

Overall, I find the main idea of this work interesting and valuable. As I mentioned in **strengths** calibrating the robustness of adversarially trained models is an appealing line of work. However, due to the confusing presentation of results, where TRADES seems to always lie on the same Pareto front as thee proposed models, and lack of results on the standard $\epsilon$ regimes I have my concerns about the credibility of the results. At this stage, I am voting for rejection, but I strongly urge the authors to engage with my questions during the rebuttal. I might have not understood some parts of their work properly, so if they can clarify my concerns, I will be willing to increase my score significantly.

---

### Official Review · Reviewer_HNzQ · 2022-10-24

**Confidence:** 4
**Correctness:** 4
**Technical Novelty And Significance:** 3
**Empirical Novelty And Significance:** 2
**Recommendation:** 5

**Clarity, Quality, Novelty And Reproducibility:**

The presentation of this paper could be improved:
* It would help to clarify in the abstract (maybe even in the title) that this paper is about adversarial robustness.
* I disagree with inserting Table 1 in the introduction. This table very misleading. It presents results without proper context or consideration for baselines. Emphasizing the improvement in robust accuracy over a standard neural network makes it seem like this is the first work that improves robust accuracy.
* The shapes are barely visible in Figure 1.
* Both the abstract and the introduction mention the 'compositional architecture' but its only explained in the methods section what this term refers to. Please clarify it in both the introduction and the abstract.
* I am not convinced that the best category for this paper is Social Aspects of ML. There are no social aspects considered in this work. Is there another category that would be a better fit for it?
* The caption of Figure 2 does not explain what the different symbols are. It mentions different $\beta$s, but does not explain what they are. Figure 3 caption is a bit better, but still difficult to understand.
* The description of robust selection is a bit confusing. It would be better to start with explaining the intuition. It should clarify that robust selection measures the percentage of inputs that are consistently selected for robust classification after perturbation.

Novelty:
* The paper does a very good job at citing related works and comparing against them.
* The idea of minimizing robust inaccuracy and make the deferral based on this is exciting and it is novel to my knowledge.

Reproducibility:
* I did not examine the supplied code, but I don't expect that it would be difficult to reproduce the results.


**Strength And Weaknesses:**

The proposed method is well motivated and it seems to offer a competitive tradeoff between the objectives. It very nice that the paper looks both at empirical robustness as well as certified robustness. They are sightly different takes on the same objective, and the method seems to work similarly well for both.

The efficacy is demonstrated in a few experiments. The experiments are thorough and good quality in my opinion.

First, it is shown that the robust network is good at reducing the robust inaccuracies, although often at the cost of slightly reduced robust accuracy (Figure 2).

Then, it is shown that the robustness criteria leads to abstaining less often while having higher robust accuracy at the same time (Figure 3). I think it's particularly interesting to see that abstaining based on stability works better than explicitly trained deferral systems and I am curious to hear the author's thoughts on why this happens. Table 2 presents similar results to Figure 3 for certified robustness, but these are not as conclusive. For the (Schwag et al.) model, $L_{CRA}$ has a clear edge, but for the (Cohen et al.) model, $L_{CRA}$ only wins in one of the two competing objectives.

Finally, Figure 4 shows robust accuracy vs natural accuracy (i.e. accuracy). These results suggest that the proposed method offers a competitive tradeoff, but it is not better both in robust accuracy and natural accuracy than the baseline TRADES. Although admittedly, TRADES is a very strong baseline tuned for the CIFAR10 dataset. It's important to emphasize, however, that the model does very well compared to other high natural accuracy approaches.

Overall, we see that the proposed method often does not beat the baselines in all objectives, rather, it offers a competitive tradeoff between them. It performs the best when compared to models with high natural accuracy.

A weakness is that the paper does not go far in discussing the limitations: what is the computational cost? what happens when the threat $\epsilon$ is under/overestimated?

**Summary Of The Paper:**

The paper proposes a training approach that optimizes the adversarial robustness of a neural network without hurting prediction accuracy.

The proposed methodology has two key components. First a network is trained to maximize robust accuracy (number of examples where the input is correctly classified and this prediction is stable in a small perturbation region) while minimizing robust inaccuracy (number of examples where the prediction is stable, but wrong). Then, this is combined with a traditionally trained network. When the prediction of the robust network is stable, then it is used for prediction. Otherwise the prediction is deferred to the traditionally trained network.

This configuration achieves the best of both worlds. High robust accuracy with high overall accuracy.

**Summary Of The Review:**

The paper is borderline in my opinion. The idea is interesting and novel, but it is only supported empirically and the experiments show that it often does not beat the baselines, it offers a tradeoff between the objectives. The presentation could be improved.

---

### Official Review · Reviewer_6trG · 2022-10-25

**Confidence:** 4
**Correctness:** 3
**Technical Novelty And Significance:** 3
**Empirical Novelty And Significance:** 3
**Recommendation:** 3

**Clarity, Quality, Novelty And Reproducibility:**

The idea seems novel and is worth exploring. But the clarity, quality and presentation needs to be improved.


**Strength And Weaknesses:**

While the proposed technique is interesting, I believe the paper is a little bit under-cooked and needs a significant amount of work to be published. Here are some key concerns I have:

* Experiments:

    - Why isn't the proposed technique used to train models from scratch? Is it because of computational challenges? The current evaluation mechanism of 'fine-tuning existing models' makes it harder to understand why the proposed algorithm is working the way it is. I would appreciate if the authors train their models from scratch and compare with standard adversarial training techniques like AT, TRADES.

    - It looks like the existing models were trained with epsilon=8/255, whereas the fine-tuning is done with smaller epsilon's. Why is there this mismatch? Can the authors report numbers for the setting where the epsilon used to train existing model is same as the epsilon used in fine-tuning?

    - In section 7.1, it is claimed that the proposed technique gives Pareto front of optimal solutions. How can this claim be made just based on Figure 2?

    - The curves in Figure 2 are confusing. Why are the triangles (TRADES) connected with circles (ERA)?

    - For models fine-tuned using TRADES, can we vary the regularization parameter lambda and obtain the robust accuracy vs robust inaccuracy curves curves as in Figure 2? How do the curves compare with the results in Figure 2?

    - I have concerns regarding "robust selection" described in Section 5.  For the "robustness indicator selector" to be robust, it looks like we need the model to be robust on double the region (2*epsilon). But the model is trained to be robust only for perturbations of size epsilon. Given this mismatch, one would expect the robustness indicator to be always 0 when there are adversarial perturbations.  Consequently, the model should always abstain. But I don't see this in the experimental results in sections 7.2, 7.3. Why is this the case? This also makes me wonder if one can come with a better attack (than APGD) to break the proposed technique. Let's say the attacker carefully chooses a point such that it is at most epsilon away from the original image, and at the same time the point is such that the model's predictions are not constant in an epsilon ball around it. Such a point would make the model to abstain and lead to incorrect predictions and make the model less robust. Is this correct? It'd be great if the authors comment on this.

   - At a number of places in the paper, it is claimed that the proposed technique improves the robustness by 61%. I find these claims to be incorrect and misleading. When I look at the results, I don't see such huge gap in performance between baselines such as TRADES and the proposed approach.

 * The clarity and presentation of the paper needs to be significantly improved. I had to take multiple passes through the paper to clearly understand the terminology, notation and the main message. For example, 'robust inaccuracy' is used in multiple places in the introduction, but is never defined. I have to go to section 3 to understand it. No background was provided on certified robustness, TRADES. Only a very few people working in this area might understand this paper. In addition, the figures are hard to read. It is not clear what improvements the proposed technique is achieving. There are no error bars. So the statistical significance of the results is questionable.

* There are several works that address the robustness vs accuracy trade-off. But these techniques were never properly compared with the proposed technique (neither experimentally nor methodologically). So it is hard to understand how good the proposed technique is and what additional insights it brings over existing techniques.

**Summary Of The Paper:**

The paper addresses the robustness vs accuracy trade-off that is often observed in adversarial training of neural networks. It proposes the following ensembling technique to overcome this trade-off. The ensemble is made up of two models. The first model is trained using standard adversarial robustness techniques with one key modification: it is trained to be robust only on points that it accurately classifies. On points that it incorrectly classifies, the model is encouraged to be non-robust. The second model in the ensemble is a non-adversarially trained model (i.e, model trained to minimize standard classification error). These two models are aggregated as follows: any input is first sent through the first model. If the first model chooses to abstain, it is sent to the second model whose output is taken as the final output. If the first model doesn't abstain, then the output of the first model is considered to be the final output.

**Summary Of The Review:**

While the proposed approach is interesting, it is presented poorly, and not thoroughly compared with baselines. Moreover, the work isn't placed properly among the long line of work on robustness vs accuracy trade-off.  It'd be great if the authors address my concerns above.

---

### Decision · Program_Chairs · 2023-01-20

**Decision:**

Reject

**Justification For Why Not Higher Score:**

The paper is far below the acceptance threshold.


**Justification For Why Not Lower Score:**

N/A


**Metareview: Summary, Strengths And Weaknesses:**

Improving robustness without sacrificing accuracy is a very important topic in machine learning for both its theoretical and practical significance. The motivation for addressing this issue is well articulated in the paper. Nevertheless, our general concern is that this work has not been put in the right perspective on multiple aspects as detailed in our review comments. Among other things, this may mislead the reader to overweigh the significance of this work, as summarized by certain claims. While we agree that the work may have potential to make its own contributions, significant revision is deemed necessary. We hope the authors find our comments and suggestions useful for them to revise their paper.